# Cold protection allows local cryotherapy in a clinical-relevant model of traumatic optic neuropathy

Yikui Zhang[1]*, Mengyun Li[1†], Bo Yu[1†], Shengjian Lu[1†], Lujie Zhang[2†], Senmiao Zhu[1†], Zhonghao Yu[1†], Tian Xia[1†], Haoliang Huang[3], WenHao Jiang[1], Si Zhang[1], Lanfang Sun[1], Qian Ye[1], Jiaying Sun[1], Hui Zhu[1], Pingping Huang[1], Huifeng Hong[1], Shuaishuai Yu[4], Wenjie Li[2], Danni Ai[2], Jingfan Fan[2], Wentao Li[5], Hong Song[5], Lei Xu[6], Xiwen Chen[7], Tongke Chen[7], Meng Zhou[8], Jingxing Ou[9,10], Jian Yang[2]*, Wei Li[11]*, Yang Hu[3]*, Wencan Wu[1]*

[1]The Eye Hospital, School of Ophthalmology & Optometry, Wenzhou Medical University, Wenzhou, China; [2]Beijing Engineering Research Center of Mixed Reality and Advanced Display, School of Optics and Photonics, Beijing Institute of Technology, Beijing, China; [3]Department of Ophthalmology, Stanford University School of Medicine, Palo Alto, United States; [4]School of Laboratory Medicine and Life Sciences, Wenzhou Medical University, Wenzhou, China; [5]School of Computer Science & Technology, Beijing Institute of Technology, Beijing, China; [6]Medical Radiology Department, 2nd Affiliated Hospital, Wenzhou Medical University, Wenzhou, China; [7]Animal Facility Center, Wenzhou Medical University, Wenzhou, China; [8]School of Biomedical Engineering, The Eye Hospital, School of Ophthalmology & Optometry, Wenzhou Medical University, Wenzhou, China; [9]Department of Hepatic Surgery and Liver Transplantation Center of the Third Affiliated, Hospital, Guangdong Province Engineering Laboratory for Transplantation Medicine, Guangzhou, China; [10]Guangdong Key Laboratory of Liver Disease Research, the Third Affiliated Hospital of Sun Yat-sen University, Guangzhou, China; [11]Retinal Neurophysiology Section, National Eye Institute, National Institute of Health, NIH, Bethesda, United States

*For correspondence:
zhang.yikui@wmu.edu.cn (YZ);
jyang@bit.edu.cn (JY);
liwei2@nei.nih.gov (WL);
huyang@stanford.edu (YH);
wuwencan@wmu.edu.cn (WW)

†These authors contributed equally to this work

Competing interest: The authors declare that no competing interests exist.

**Abstract:** Therapeutic hypothermia (TH) is potentially an important therapy for central nervous system (CNS) trauma. However, its clinical application remains controversial, hampered by two major factors: (1) Many of the CNS injury sites, such as the optic nerve (ON), are deeply buried, preventing access for local TH. The alternative is to apply TH systemically, which significantly limits the applicable temperature range. (2) Even with possible access for 'local refrigeration', cold-induced cellular damage offsets the benefit of TH. Here we present a clinically translatable model of traumatic optic neuropathy (TON) by applying clinical trans-nasal endoscopic surgery to goats and non-human primates. This model faithfully recapitulates clinical features of TON such as the injury site (pre-chiasmatic ON), the spatiotemporal pattern of neural degeneration, and the accessibility of local treatments with large operating space. We also developed a computer program to simplify the endoscopic procedure and expand this model to other large animal species. Moreover, applying a cold-protective treatment, inspired by our previous hibernation research, enables us to deliver deep hypothermia (4 °C) locally to mitigate inflammation and metabolic stress (indicated by the transcriptomic changes after injury) without cold-induced cellular damage, and confers prominent neuroprotection both structurally and functionally. Intriguingly, neither treatment alone was effective, demonstrating that in situ deep hypothermia combined with cold protection constitutes a breakthrough for TH as a therapy for TON and other CNS traumas.

## Editor's evaluation

This manuscript describes a new surgical procedure to access the optic nerve in large mammals to provide therapeutic hypothermia. Therapeutic hypothermia is a powerful idea to prevent degeneration of the nervous system following trauma or other insults. This study represents a significant step forward in the use of therapeutic hypothermia in the treatment of ocular conditions.

## Introduction

Therapeutic hypothermic (TH) has a long history and was firstly described in the Edwin Smith papyrus over 5000 years ago (*Wang et al., 2006*). Although TH has shown neuroprotection in animal models of central nervous system (CNS) trauma such as spinal cord injury (SCI) and traumatic optic neuropathy (TON) by reducing neuroinflammation and alleviating metabolic demand (*Martirosyan et al., 2017*; *Rey-Funes et al., 2017*), its clinical application following CNS injury remains controversial. Randomized controlled clinical trials of TH for traumatic brain injury (TBI) failed to show beneficial effect (*Clifton et al., 2001*; *Clifton et al., 2011*; *Kramer et al., 2012*). In SCI, TH has not been proven to be neuroprotective in randomized controlled clinical trials, and its effect in animal models is inconsistent (*Martirosyan et al., 2017*).

At least two major factors hamper the clinical application of TH in CNS trauma. First, many CNS injury sites are located deeply, preventing access for local TH. Instead, TH is applied systemically. But systemic drops in body temperature increase the risk of cardiopulmonary disorders, coagulopathy, and electrolyte disturbances (*Schubert, 1995*), so systemic TH is mainly limited to 34°C and 36°C. The limited temperature range of TH may account for its inadequate efficacy. Secondly, even if CNS trauma is accessible for local deep hypothermia, the cold destroys neuronal microtubules and may counteract the benefit of TH (*Ou et al., 2018*). Our previous study discovered a cold-protective mechanism in hibernators, and found that cold-protective reagents such as protease inhibitors (PI) rescue cold-induced cell damage. If these reagents were applied during local TH, then TH would be a more precise, safe, and effective tool to treat CNS traumas (*Ou et al., 2018*).

The optic nerve (ON), which collects axons from retinal ganglion cells (RGCs), is an ideal CNS tissue to study CNS trauma because it is structurally simple.However, its deep location in the orbit and skull base hampers preclinical tests of local treatment for optic neuropathy (*Figure 1A*). Minimally invasive trans-nasal endoscopy is widely used in modern neurosurgeries to access the distal ON (pre-chiasmatic and chiasmatic ON) in case of tumor or optic canal fracture (*Figure 1B*; *Chamoun and Couldwell, 2011*; *Casler et al., 2005*; *Goudakos et al., 2011*; *Yan et al., 2017*). The large sphenoid sinus along the distal ON provides sufficient space for local treatment (*Figure 1A–C*). Here, we back-translated clinical trans-nasal endoscopy in large animals to safely expose the distal ON, allowing local damage, observation, and modulation. To facilitate clinical translation, we established a de novo TON model in goats and non-human primates by performing pre-chiasmatic ON crush. These models recapitulated clinical features of TON such as the injury site, the time course of progression, and the capability of local treatment via trans-nasal endoscopy.

To serve the twin goals of administrating local deep cooling therapy and preventing cold-induced microtubule instability, we applied both TH and cold-protective reagents locally at the injured pre-chiasmatic ON, and found this noval local treatment achieved significant structural and functional neuroprotection. Next, we developed a computer program to detail feasible surgical pathways and to optimize the size of endoscopic tools according to a CT scan of the skull. This program was successful in goats, minipigs, beagles, and rhesus macaques. We trust that this software will help other research groups to replicate our novel TH strategy, and will facilitate preclinical tests for other local TON treatments.

This novel combination of TH and a cold-protection strategy can be readily applied not only in TON patients using the same procedures and devices, but also can potentially revolutionize conventional TH for other CNS trauma such as SCI and TBI. Furthermore, this translatable TON model allows preclinical testing of local treatments for axonopathy. Therapeutic approach for TON may move from less effective strategies (for example, observation alone, or systemic medication; *Yu-Wai-Man and Griffiths, 2013*) towards more targeted and potentially more effective treatment *via* minimally invasive trans-nasal endoscopy.

**eLife digest** Hypothermic therapy is a radical type of treatment that involves cooling a person's core body temperature several degrees below normal to protect against brain damage. Lowering body temperature slows blood flow, which reduces inflammation, and eases metabolic demands, similar to hibernation. It can also reduce lasting damage to the brain and aid recovery when used to treat people who have gone into cardiac arrest, where their heart suddenly stops beating.

Recently, there has been renewed interest in using hypothermic therapy to treat people who have sustained traumatic brain injuries, which can cause brain swelling, and other nerve injuries. However, its use remains controversial because clinical trials have failed to show that inducing mild hypothermia provides any benefit for people with severe nerve injuries. This might be because cooling cells to near-freezing temperatures can damage their internal structural supports, called microtubules, thwarting any therapeutic benefit.

Traumatic optical neuropathy is a type of injury in which the optic nerve – the nerve that connects the eyes to the brain – is damaged or severed, causing vision loss. There is currently no clinically proven treatment for this condition, nor is there a system that can test local treatments in large animals as a prior test to using the treatment in the clinic. Therefore, Zhang et al. wanted to establish such a animal model and test whether local hypothermic therapy could help protect the optic nerve.

Zhang et al. used a surgical tool guided by an endoscope (a thin plastic tube with a light and camera attached to it) to injure the optic nerves of goats, and then deliver hypothermic therapy. To cool the surgically-injured nerves to a chilly 4C, Zhang et al. applied a deep-cooling agent, using a second reagent (a cocktail of protease inhibitors) to protect the cells' microtubules from cold-induced damage, an insight gained from a previous study of hibernating animals. This was critical, as the hypothermic therapy was only effective when the secondary protective agent was applied. The combination therapy developed by Zhang et al. relieved some aspects of nerve degeneration at the injury site and activated an anti-inflammatory response in cells, but did not restore vision.

To simplify surgical techniques, Zhang et al. also developed a computer program which generates virtual surgical paths for up-the-nose endoscopic procedures based on brain scans of an animal's skull. This program was successfully applied in a range of large animals, including goats and macaque monkeys.

Zhang et al.'s work establishes a method to study treatments for traumatic optical neuropathy using large animals, including hypothermic therapy. The methods developed could also be useful to study other optic nerve disorders, such as optic neuritis or ischemic optic neuropathy.

## Results

### Goat is an advantageous species for trans-nasal endoscopic access of the chiasmatic and pre-chiasmatic ON

We first looked for animal species suitable for trans-nasal exposure of the chiasmatic and pre-chiasmatic ON. Skull computerized tomography (CT) scan showed that the frontal cortex of the domestic pig and beagle (*Figure 1D*, asterisk) falls between the nasal cavity (*Figure 1D*, number sign) and the pre-chiasmatic ON (*Figure 1D*, blue rectangle), thus blocking surgical access. The narrowest portion of rhesus macaque's sphenoid bone body (*Figure 1D*, red circle) is too narrow (around 2.5 mm in width) to allow a conventional endoscopic microdrill (2.9 mm in diameter) to pass through. On the other hand, the size of the goat's sphenoid bone body (13 mm in width) is similar to that of the human's sphenoid sinus (20 mm in width), and there is no cortex in between the nasal cavity and the pre-chiasmatic ON (*Figure 1D*). Additionally, Saanen goats are readily available, easy to raise and handle, genetically editable (*Kalds et al., 2019*), and have a genome that is well studied (*Araújo et al., 2006*; *Martin et al., 2018*; *Gipson, 2019*). Therefore, we decided to use Saanen goat as our research animal model.

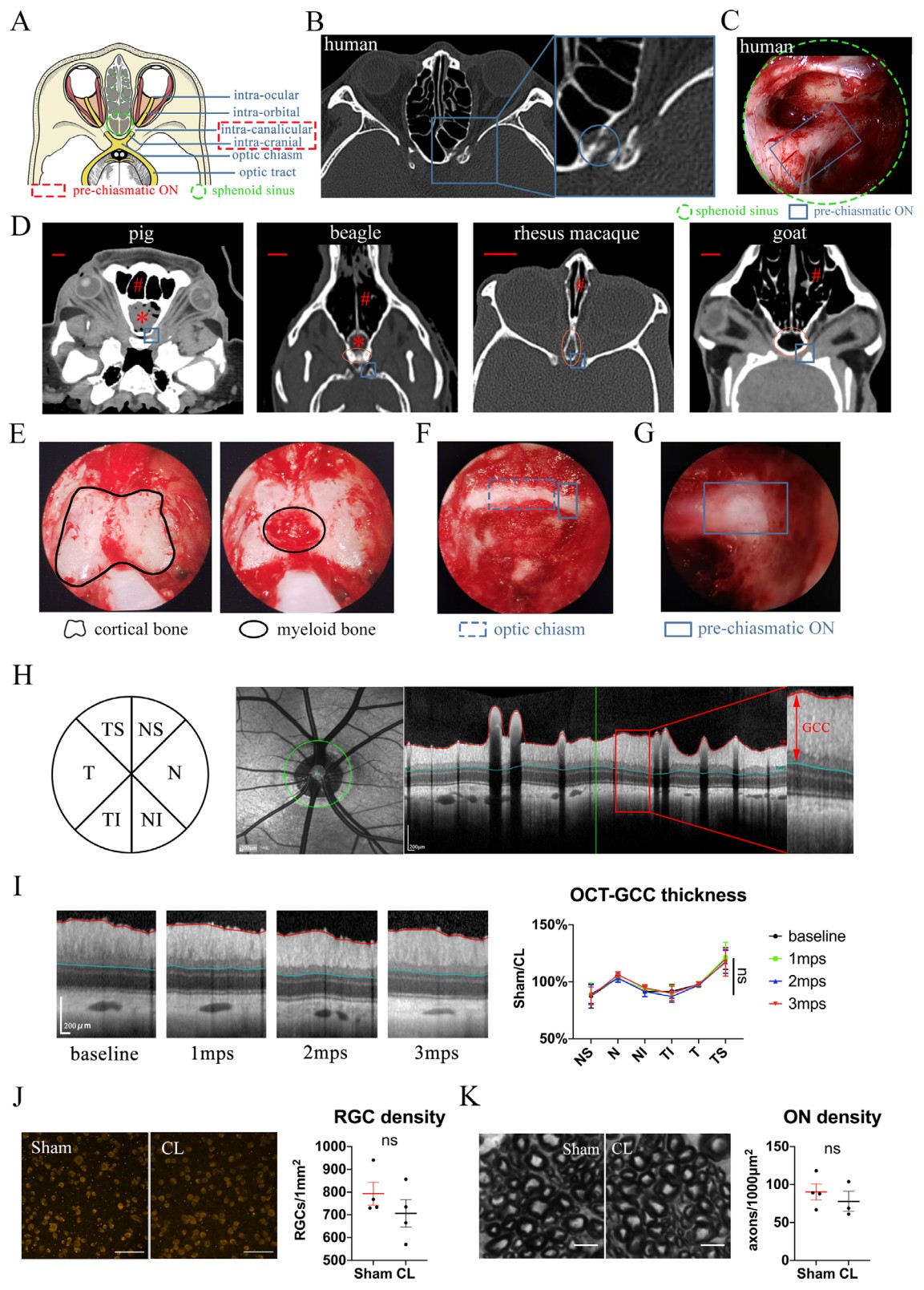

**Figure 1.** Trans-nasal endoscopic-mediated exposure of pre-chiasmatic ON in goat is feasible and safe. (**A**) Scheme of anatomic segments of ON and sphenoid sinus (green circle) in human. The pre-chiasmatic ON was shown in the dashed box. (**B**) CT scan of TON patient with optic canal fracture (blue circle). (**C**) Endoscopic view of human's pre-chiasmatic ON (blue rectangle) within the large sphenoid sinus (shown in the green circle). (**D**) Representative skull CT scans of pre-chiasmatic ON (blue rectangle) in pigs, beagles, rhesus macaques and goats. Number sign indicated the nasal cavity. Asterisk

*Figure 1 continued on next page*

*Figure 1 continued*

indicated the dropping frontal cortex in pigs and beagles. Red circle indicates the sphenoid bone body in beagles, rhesus macaques and goats. Scale bar = 10 mm. (**E**) Endoscopic image of a goat's sphenoid bone body, which is made of cortical bone (left panel) and trabecular bone with myeloid tissue (right panel). (**F**) Endoscopic image of a goat's chiasmatic ON (blue dashed rectangle) and pre-chiasmatic ON (blue rectangle) with bony wall unremoved. (**G**) Endoscopic exposure of a goat's pre-chiasmatic ON (blue rectangle) with its anterior bony wall removed. (**H**) Illustration of GCC thickness measurement by OCT retinal imaging around the optic nerve head in six regions (T: temple, N: nasal; S: superior; I: inferior). (**I**) Representative OCT images of the eye with its pre-chiasmatic ON exposed (sham surgery eye) (left panel) and quantification of GCC thickness ratio of the sham eyes to the contralateral naïve eyes before and after pre-chiasmatic ON exposure. n = 4. Scheirer-Ray-Hare test with Dunn's multiple comparison (compared with the baseline). (**J**) Representative immunostaining images of the RBPMS positive RGCs in the retinal flat-mounts (left panel) and quantification of RGCs densities of the sham eyes and the contralateral eyes at 3 months after pre-chiasmatic ON exposure (right panel). n = 4. Wilcoxon test. Scale bar = 100 µm. (**K**) Representative microscopic images of semi-thin cross sections of ON (left panel) and quantification of axonal densities of the sham eyes and the contralateral eyes at 3 months after pre-chiasmatic ON exposure (right panel). n = 3–4. Unpaired t-test. Scale bar = 5 µm. Data were presented as mean ± s.e.m. ns: p > 0.05, not significant. CL: contralateral, mps: month post-surgery (post sham surgery). The source data is in '*Figure 1—source data 1*'.

The online version of this article includes the following source data and figure supplement(s) for figure 1:

**Source data 1.** Endoscopic exposure of pre-chiasmatic ON is safe in goats.

**Figure supplement 1.** Endoscopic exposure of pre-chiasmatic ON is safe in goats.

**Figure supplement 1—source data 1.** Source data for *Figure 1—figure supplement 1*.

## Trans-nasal endoscopic exposure of the pre-chiasmatic on in goats is feasible and safe

By using clinically available trans-nasal endoscopes (*Video 1*), we exposed the goat's sphenoid bone body (*Figure 1E*), and then removed its anterior cortical bone and inner bone marrow to create an artificial sphenoid sinus. The anterior bony wall of the chiasmatic and pre-chiasmatic ON laid posteriorly in the sinus (*Figure 1F*). We then exposed the pre-chiasmatic ON by drilling off the anterior bony wall (*Figure 1G*, *Video 2*). In this initial procedure, the exposed ON was not further manipulated and is henceforth denoted 'sham'.

To evaluate the safety of trans-nasal endoscopic exposure of the pre-chiasmatic ON, we followed the animal for up to 3 months after surgery. The thickness of the ganglion cell complex (GCC, including RGCs' axons, somas and dendrites) in six areas around the optic nerve head was measured by optical coherence tomography (OCT) at different time points (*Figure 1H*). There was no significant change in GCC thickness over 3 months after pre-chiasmatic ON exposure in either the sham eyes (*Figure 1I*) or the contralateral naïve eyes (*Figure 1—figure supplement 1A*). In line with the OCT results, both the RGC somal densities and axonal densities were unchanged in the sham eyes compared to the contralateral naïve eyes at 3 months post surgery (3 mps) (*Figure 1J and K*). Consistent with morphological studies, functional readouts (pupillary light reflex (PLR), pattern electroretinogram (PERG), flash visual evoked potential (FVEP) tests) of the sham eyes were not changed over 3 mps compared with the baseline (*Figure 1—figure supplement 1B*, **C, D**). Taken together, these results show that endoscopic exposure of the pre-chiasmatic ON is feasible and safe.

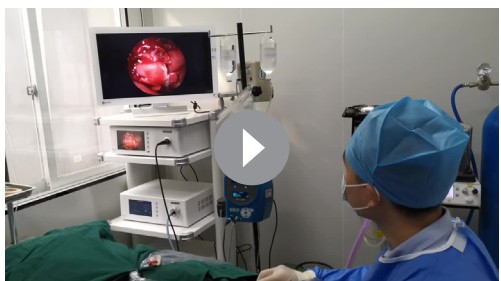

**Video 1.** Surgical settings for trans-nasal endoscopic exposure of the pre-chiasmatic optic nerve in a goat.
https://elifesciences.org/articles/75070/figures#video1

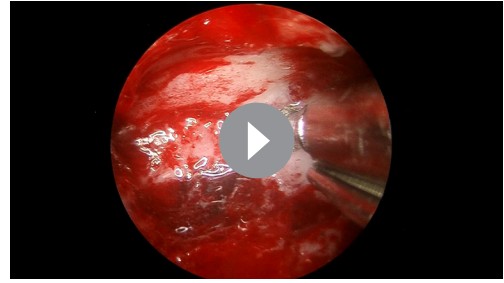

**Video 2.** Exposure of the pre-chiasmatic optic nerve via trans-nasal endoscopy in a goat.
https://elifesciences.org/articles/75070/figures#video2

## Trans-nasal endoscopy-mediated pre-chiasmatic on crush causes visual loss in goat

Most human TON occur in the pre-chiasmatic ON due to force transmission (*Figure 1B*; *Yan et al., 2017*; *Yu-Wai-Man, 2015*). To establish a clinically relevant animal model of TON which enables local modulation of the injured pre-chiasmatic ON, we performed pre-chiasmatic ON crush in goat using a trans-nasal endoscopic approach. As illustrated in *Figure 2A*, we first drilled the anterior bony wall to expose the pre-chiasmatic ON, and then manually compressed the pre-chiasmatic ON with the blunt blade of a periosteal elevator until it met the optic canal's posterior bony wall (*Figure 2B*, *Video 3*).

The visual function of the injured eye was lost after pre-chiasmatic ON crush. Both the direct PLR (dPLR) in the injured eyes and the indirect PLR (iPLR) in the contralateral control eyes disappeared after injury (*Figure 2C*), whereas the dPLR in the contralateral eyes and the iPLR in the injured eyes remained intact (*Figure 2—figure supplement 1A*, **left panel**). These results suggested that ON function in the injured eyes was lost whereas the contralateral ON was still functional; additionally, the oculo-motor nerves of both eyes were unaffected by pre-chiasmatic ON crush (*Kerrison et al., 2001*). Consistently, the ratio of the FVEP P1-N1 and P2-N1 amplitudes of the injured eyes to the contralateral eyes decreased significantly compared to the baselines (*Figure 2D*), indicating impairment of the retino-geniculate pathway (*Odom et al., 2010*). Interestingly, the ratio of PERG P1-N1 amplitude of the injured eyes to that of the contralateral eyes remained unchanged at 1 week post injury (1 wpi), then decreased significantly at 1, 2, and 3 months post injury (mpi) compared to the baseline (*Figure 2E*), indicating that RGC function degraded over time (*Porciatti, 2015*).

Taken together, the visual deficits (PLR, PERG and FVEP abnormality) presented in the TON model are similar to the clinical manifestations of patients with monocular TON (that is, Marcus Gunn pupil, loss of visual acuity, and reduction in VEP signals) (*Singman et al., 2016*).

## Progressive RGC and on degeneration in goat TON model

The GCC thickness ratio of the injured eyes to contralateral control eyes remained stable at 1 wpi, then decreased progressively over 3 mpi (about 89%, 81%, 74% of the contralateral eyes at 1, 2, 3 mpi, respectively) (*Figure 3A*). The GCC thickness in the contralateral eyes was unchanged after ON crush (*Figure 2—figure supplement 1A*, **right panel**). Progressive GCC thinning occurred in all six areas around the ON head (*Figure 3A*), indicating that ON crush caused widespread axonal damage with no quadrant being spared. Importantly, the time course of GCC thinning in our model closely resembles that of TON patients (*Miyahara et al., 2003*; *Shi et al., 2013*).

There was no significant RGC soma loss at 1 mpi, yet the RBPMS (RNA-binding protein with multiple splicing) immunoreactivity in some RGCs was weak and irregular (*Figure 3B*). At 3 mpi, only 25% of RGCs survived in the injured eyes compared with the contralateral eyes (*Figure 3B*). RGC density in the contralateral eyes remained stable at 1 and 3 mpi (*Figure 2—figure supplement 1B*, **left panel**). It is worth noticing that our previous study in goat showed that retrobulbar ON crush resulted in more severe RGC loss at 3 mpi (less than 5% survival) (*Zhang et al., 2020*).

To study the temporal and spatial patterns of axonal degeneration after pre-chiasmatic ON crush, we examined semi-thin cross-sections from four different sites of the ONs (named #1, 2, 3, 4) of both eyes at 1 and 3 mpi (*Figure 3C*). For the contralateral ON, the surviving axonal densities remained statistically the same at 1 and 3 mpi, and there was no significant difference in axonal densities among different ON sites (*Figure 2—figure supplement 1B*, **right panel**). For the injured eyes, there were few surviving axons at distal regions (#3, #4) at 1 and 3 mpi. In contrast, axonal densities of the proximal ON segments (#1, #2) of the injured eyes were almost intact at 1 mpi and then decreased dramatically at 3 mpi (*Figure 3D–F*), indicating progressive retrograde axonal degeneration, which was also observed in rodent models of distal ON injury (*Richardson et al., 1982*).

Taken together, these morphological studies demonstrate progressive retrograde neural degeneration in our TON model, the time course of which is similar to that of clinical TON.

## Transcriptomic analysis revealed early changes in ischemia, inflammation, and metabolic pathways at the injured optic nerve

To explore early changes and identify potential therapeutic targets in the injured eye, we sampled from the retina (R), proximal (#1) and pre-chiasmatic (#2) ON segments in both the injured eye and the contralateral eye at 1 day post injury (dpi), and performed RNA-sequencing (*Figure 4A*). Compared

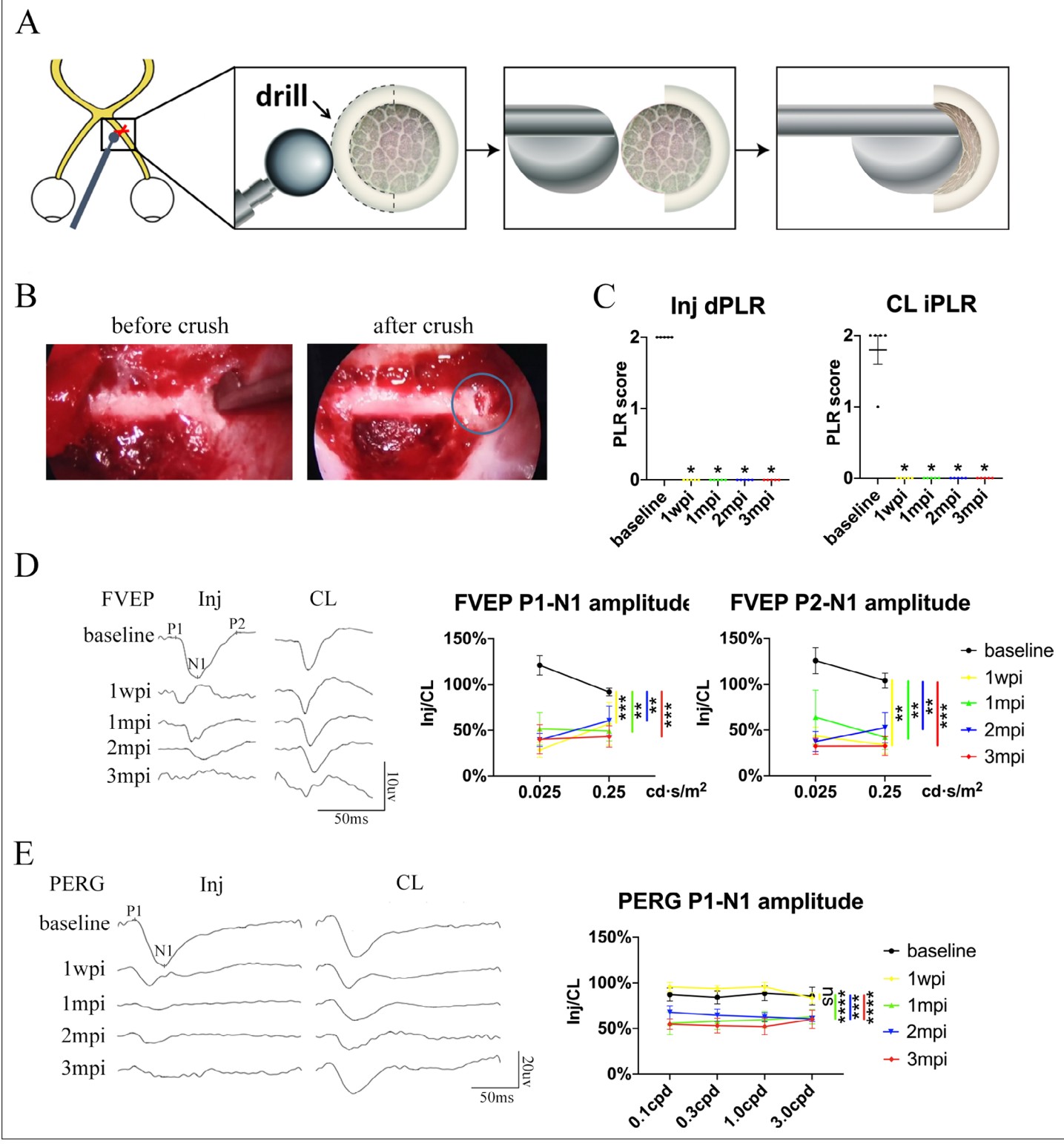

**Figure 2.** Trans-nasal endoscopy-mediated pre-chiasmatic ON crush in goat leads to loss of visual function. (**A**) Schematic illustration of pre-chiasmatic ON exposure and crush injury. (**B**) Endoscopic views before (left panel) and after pre-chiasmatic ON crush (right panel). Crush site was shown in the blue circle. (**C**) Loss of direct pupillary light response (dPLR) in the injured eyes (left panel) and indirect PLR (iPLR) in the contralateral eyes (right panel) after crush. Friedman test with Dunn's multiple comparison (compared with baseline), p = 0.0497. (**D**) Representative FVEP waveforms of the injured eye and the contralateral eye at light intensity of 0.025 cd·s/m² (left panel) and quantification of FVEP P1-N1 and P2-N1 amplitudes ratios of the injured eyes to the contralateral eyes (right panel) at different time points before and after pre-chiasmatic ON crush. Left panel: Two-way ANOVA with Tukey's multiple

*Figure 2 continued on next page*

*Figure 2 continued*

comparison. Right panel: Scheirer-Ray-Hare test with Dunn's multiple comparison (compared with the baseline). (**E**) Representative PERG waveforms of the injured eye and contralateral eye at spatial frequency of 0.1 cpd (left panel) and quantification of PERG P1-N1 amplitude ratios of the injured eyes to the contralateral eyes at spatial frequencies of 0.1, 0.3, 1.0, 3.0 cpd (right panel) before and after pre-chiasmatic ON crush. Two-way ANOVA with Tukey's multiple comparison (compared with the baseline). Data were presented as mean ± s.e.m, n = 5. ns: not significant, * p < 0.05, ** p < 0.01, *** p < 0.001, **** p < 0.0001. Inj: injured, CL: contralateral, wpi: week post-injury, mpi: month post-injury. The source data is in '*Figure 2—source data 1*'.

The online version of this article includes the following source data and figure supplement(s) for figure 2:

**Source data 1.** Source data for *Figure 2*.

**Figure supplement 1.** Contralateral eyes are intact functionally and structurally after ON crush injury.

**Figure supplement 1—source data 1.** Source data for *Figure 2—figure supplement 1*.

with the contralateral eye, there were few differentially expressed genes (DEGs) in the retina and proximal ON of the injured eye, indicating the transcriptomic status and microenvironment of the retinal and proximal axonal were stable at 1 dpi (*Figure 4B*). In addition, DEGs between the proximal and pre-chiasmatic ON of the contralateral eye were rare, suggesting that the axonal status and microenvironment were consistent between different ON segments (*Figure 4B*). In contrast, there were large amount of DEGs in the injured ON segment (Inj_#3) compared with either the contralateral pre-chiasmatic ON (ctrl_#3) or the ipsilateral proximal ON (Inj_#1). To narrow down the ranges of target genes, we extracted overlapping DEGs between Inj_#3 vs. Ctrl_#3 and Inj_#3 vs. Inj_#1, and found that 91% of the overlapping DEGs were clustered in the pathways of inflammation, ischemia, and metabolism (*Figure 4D and E*, *Figure 4—figure supplement 1A*). We further found that most of the DEGs with high connectivity ('hub genes') were also enriched in these pathways (*Figure 4—figure supplement 2A*, **B, D**). These transcriptomic changes at the injury site were unlikely intrinsic to the distal optic nerve axons and more likely occurred primarily in non-neuronal cells of the optic nerve. Gene ontology (GO) analysis of the overlapping DEGs and hub genes is listed in *Figure 4—figure supplement 1B*, *Figure 4—figure supplement 2C*.

## Protease inhibitors (PI) rescued cold-induced axonal microtubule damage

Hypothermic therapy can alleviate inflammation and reduce metabolic demand after CNS trauma (*Martirosyan et al., 2017*). However, our previous study showed that the cold destroys neuronal microtubules, and that microtubules can be rescued by cold-protective reagents such as PI (*Ou et al., 2018*). In this study, goat retinal explants cultured at 0 °C suffered severe axonal damage compared to those cultured at 37 °C, as shown by beta-3 tubulin-positive axonal length and axonal beads. When we added PI into 0 °C medium, this cold-induced axonal degradation was significantly reduced (*Figure 5A–D*).

## Trans-nasal local delivery of hypothermia and PI alleviates neural degeneration after pre-chiasmatic on crush

To prevent axons from degrading during local TH, we applied PI to the injured pre-chiasmatic ON while administering hypothermia locally (*Figure 5E and F*, *Video 4*). The temperature curves of the cooling water and the cling film over time are shown in *Figure 5G*. In the in vivo experiment, we used 4 °C, but not 0 °C, local hypothermia to the injured optic nerve. The sponge placed at the injury site was filled with goat serum with or without PI.

We found that local combinatory delivery of hypothermia and PI significantly alleviated neural degeneration compared with the no treatment group at 1 mpi in terms of PERG amplitude, GCC thickness and axonal density at the injury site (*Figure 5H–J*). In contrast, local hypothermia or

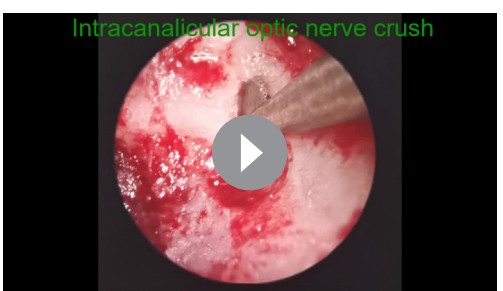

**Video 3.** Crush of the pre-chiasmatic optic nerve via trans-nasal endoscopy in a goat.
https://elifesciences.org/articles/75070/figures#video3

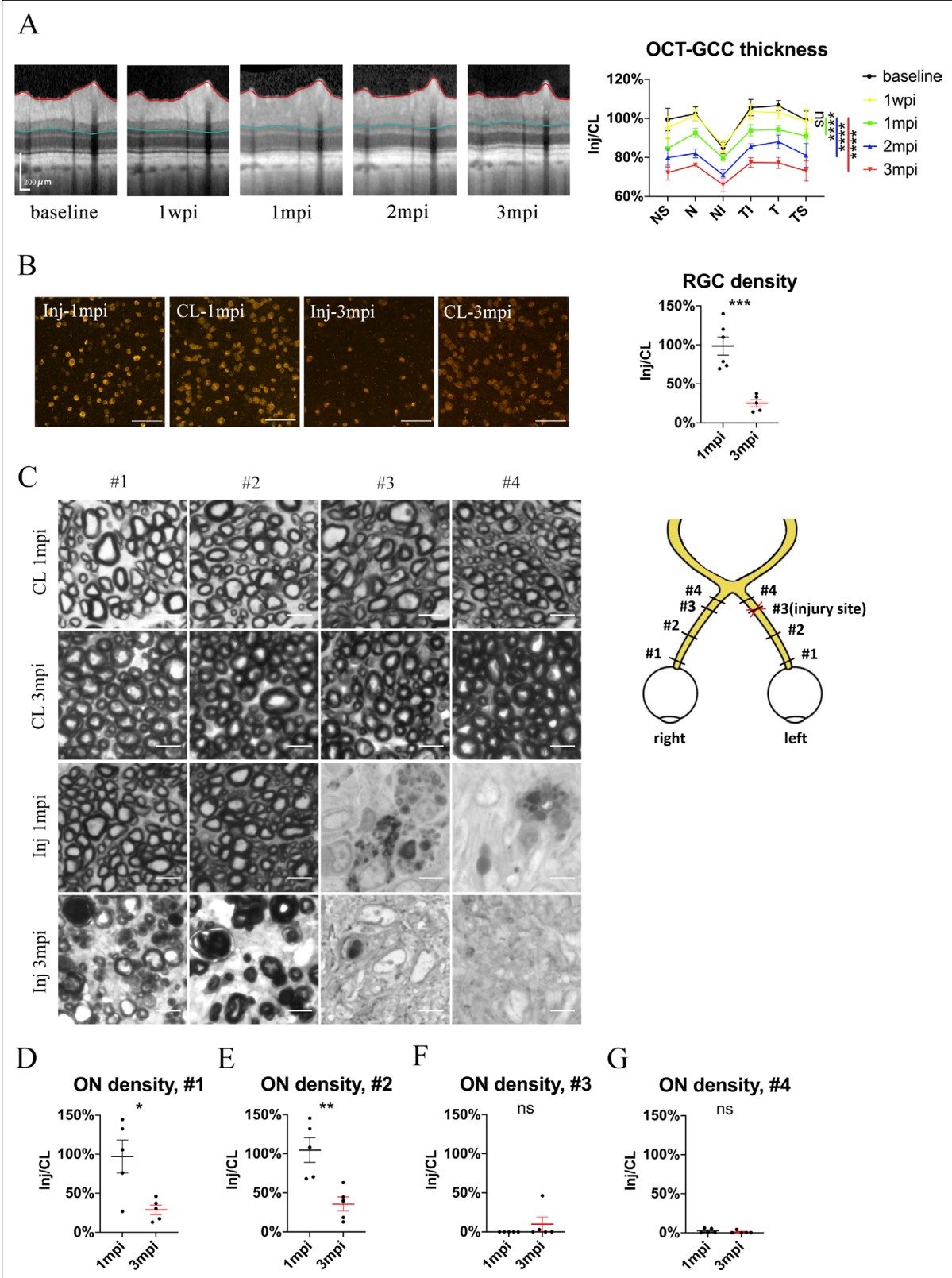

**Figure 3.** Spatiotemporal pattern of RGC and ON degeneration in goat TON model. (**A**) Representative OCT images of the injured eye (left panel) and quantification of GCC thickness ratios of the injured eyes to that of the contralateral eyes at six regions around ON head (right panel) before and after pre-chiasmatic ON crush. n = 5. Two-way ANOVA with Tukey's multiple comparison (compared with the baseline). (**B**) Representative immunostaining images of the RBPMS positive retinal ganglion cells (RGCs) in the retinal flat-mounts (left panel) and quantification of RGCs densities ratios of the injured

*Figure 3 continued on next page*

*Figure 3 continued*

eyes to the contralateral eyes at 1, 3 mpi (right panel). n = 5–6. Unpaired t-test. Scale bar = 100 μm. (**C**) Left panel: representative microscopic images of semi-thin cross sections of ON stained by PPD at different ON segments of the injured eye and its contralateral eye at 1, 3 mpi. Scale bar = 5 μm. Right panel: scheme of sampling sites at different ON segments (#1, 2, 3, 4). (**D–G**) Quantification of axonal densities ratios of the injured eyes to the contralateral eyes at different ON segments at 1 and 3 mpi. n = 5. (**D, E**) Unpaired t-test. (**E, F**) Mann-Whitney test. Data were presented as mean ± s.e.m. ns: not significant, * p < 0.05, ** p < 0.01, *** p < 0.001, **** p < 0.0001. Inj: injured, CL: contralateral, wpi: week post-injury, mpi: month post-injury. The source data is in '*Figure 3—source data 1*'.

The online version of this article includes the following source data for figure 3:

**Source data 1.** Source data for *Figure 3*.

local PI alone failed to show significant neural protection in either PERG signal or GCC thickness (*Figure 5H1*).

Neither FVEP or PLR were rescued in any treatment group (*Figure 5—figure supplement 1A-C*), since both require an intact ON. In the combinatory treatment group, there were few residual axons at the #4 ON segment (distal to the injury site) (*Figure 5—figure supplement 1D*), indicating Wallerian degeneration was not significantly rescued.

## Transcriptomic analysis of the injured optic nerve after local hypothermia/PI treatment reveals changes in immune response

To explore transcriptomic change after local treatment with deep hypothermia and PI, we sampled from the injured pre-chiasmatic ON segments at 1 day post injury (dpi), and compared transcriptomic expression between the hypothermia/PI treatment group and no treatment group. There were 264 differentially expressed genes after hypothermia/PI treatment, of which 220 genes were upregulated and 44 geneswere downregulated (*Figure 6A and B*). Gene list analysis by Metascape revealed that many inflammatory, immune-related functional cluster groups of gene ontology terms were enriched, such as neutrophil degranulation, inflammatory response, and regulation of cell adhesion (*Figure 6C*). Scored pro-inflammatory response (GO:0050729: positive regulation of inflammatory response) and anti-inflammatory response (GO:0050728: negative regulation of inflammatory response) by GSEA revealed that the enrichment score of negative regulation of inflammation was higher than that of positive regulation, indicating that hypothermia/PI treatment negatively regulated inflammatory response overall (*Figure 6D*). SsGSEA analysis (*Figure 6E*) found that the anti-inflammatory response was significantly upregulated after hypothermia treatment. QuanTIseq analysis, which was used to predict cell types within each sample, found a significant decrease in the number of neutrophils and a significant increase in the number of B cells, T cells, and mDCs (myeloid dendritic cells), indicating that innate immune response was suppressed while adaptive immune response was enhanced after hypothermia/PI treatment (*Figure 6F*). These results need to be confirmed by single-cell sequencing in the future study.

## Computer program-aided optimization of trans-nasal endoscopic surgery in multiple large animal species

To simplify the trans-nasal endoscopic surgery and expand the approach to other large animal species, we developed a computer program to generate virtual surgical paths. The flowchart of programming is detailed in *Figure 7A*. Briefly, we input CT scans of the animal's skull into the program to outline the surgical space, the non-surgical space, and the surgical target (*Figure 7B*). Then, we entered the size of the surgical microdrill to generate the surgical corridor (*Figure 7C*).

As shown in *Figure 7D*, there were five degrees of freedom for the surgical corridor when its head met the target (optic canal): rotation around x, y, z axes; spin around the body (major axis); and movement along the neck. We set the step size and ranges of rotation (around x, y, z axes) as 5° and 180°, the step size and ranges of spin as 5° and 360°, the step size and range of movement as 1 mm and 15 mm. Therefore, the total number of theoretical paths were calculated as (180/5)*(180/5)*(180/5)*(360/5)*(15/1) = 50388480. The computer program tried all these theoretical paths to find collision-free surgical corridors within the surgical space. In another words, no pixel of the surgical corridor should be included in the non-surgical space. For example, in *Figure 7E*, the surgical corridor on the right collided with the orbit (non-surgical space, shown in red), so this path was excluded. If there was no

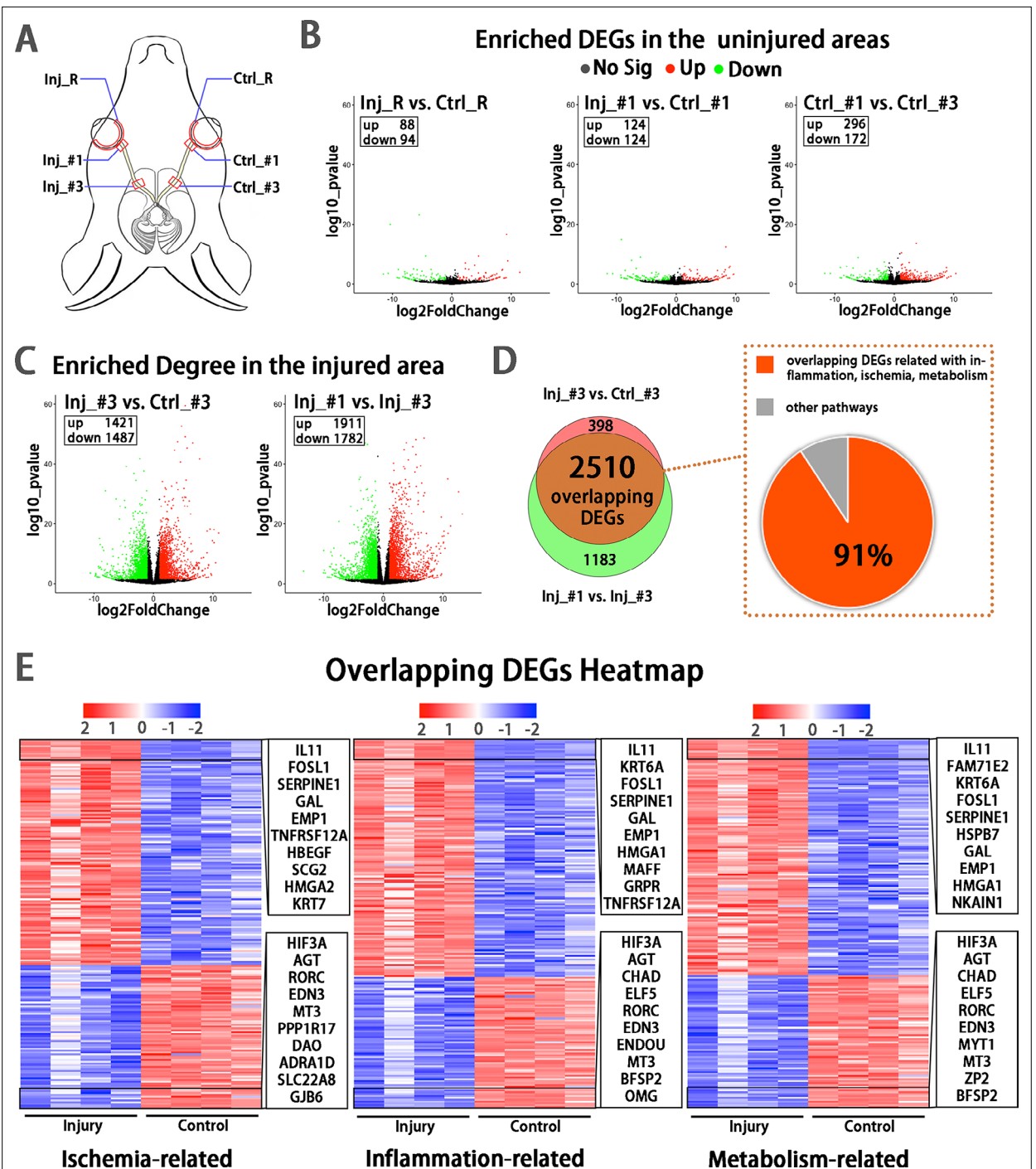

**Figure 4.** Early transcriptomic changes are confined to the injury site and mostly enriched in the pathways of ischemia, inflammation and metabolism. (**A**) Scheme of sampling sites at the retina (**R**), retrobulbar (#1) and pre-chiasmatic ON (#3) of the injury eye (Inj) and its contralateral eye (Ctrl). (**B, C**) Volcano plots showing differential expression genes (DEGs) in the non-injured areas (**B**) and the injured areas (**C**). Red dots: significant upregulated genes, green dots: significant downregulated genes, adjusted *P* < 0.05. log2FC is 1. (**D**) Venn diagram indicating the overlapping DEGs between Inj_#3 vs. Ctrl_#3 (ipsilateral-contralateral comparison) and Inj_#3 vs. Inj_#1 (proximal-distal comparison of the ipsilateral eye) (left panel), 91% of which were clustered in the pathways of inflammation, ischemia and metabolism (right panel). (**E**): Heatmap showing TOP 200 overlapping DEGs related with pathways of ischemia (left panel), inflammation (mid panel) and metabolism (right panel) based on the DEGs of Inj_#3 vs. Ctrl_#3. Ranking was determined by the magnitude of fold change. Upper box shows the top 10 up-regulated genes, and the below one shows the top 10 down-regulated genes.

The online version of this article includes the following figure supplement(s) for figure 4:

*Figure 4 continued on next page*

feasible surgical pathway with a given size of microdrill, the computer program would reduce the size of the microdrill, and repeat the searching process.

*Figure 7F–I* shows the virtual surgical paths generated by the computer program to access the pre-chiasmatic ON in goat, beagle, domestic pig and rhesus macaque, respectively. The surgery entry for each species is demonstrated on both the 3D-reconstructed skull (**left panel**) and on the horizontal plane of the CT scan (**middle panel**). The right panels demonstrate the final position of the surgical microdrill when its head meets the target (pre-chiasmatic ON in the optic canal). The step-by-step surgical path plans for goat, beagle, domestic pig and rhesus macaque are presented in *Videos 5–8*.

From the computer program, we learn that (1) there are multiple feasible surgical paths in goats (*Figure 7J*), (2) although the frontal cortex in beagles and domestic pigs seems to block access to the pre-chiasmatic ON (*Figure 1D*), it can be circumvented either inferiorly or laterally (*Figure 7G and H*), and (3) in rhesus macaques, we can access the pre-chiasmatic ON by either circumventing the narrowest portion of the sphenoid bone body (*Figure 7I*) or using a smaller surgical microdrill (2.5 mm in diameter) (*Figure 7K*).

This computer-assisted surgical path planning program can be used to (1) screen suitable animals for trans-nasal endoscopy before surgery, (2) provide a detailed roadmap for endoscopic surgery, and (3) potentially pave the way for robotic surgery to industrialize this novel TON model.

## Pre-chiasmatic TON model in rhesus macaques facilitated by computer program-aided optimization

The visual systems of non-human primates (including rhesus macaques) resemble those of humans in terms of macular structure, RGC subtypes, and ON projection pattern (*Preuss, 2004*). Therefore, we decided to expand our modeling to rhesus macaques. Directly informed by our computer program, we successfully exposed the pre-chiasmatic ON in rhesus macaque via trans-nasal endoscopy (*Video 9*), confirmed the target with the surgical navigation system (*Figure 7—figure supplement 1A*), and performed pre-chiasmatic ON crush (*Figure 7—figure supplement 1B*).

We observed substantial visual loss and neural degeneration following crush injury. The ratios of PVEP N1-P2 amplitudes and the PERG P1-N1 amplitudes of the injured eyes to the contralateral eyes fell significantly compared with baseline (*Figure 7—figure supplement 1C*, **D**). Structurally, GCC thickness in the contralateral eyes did not change over time (*Figure 2—figure supplement 1C*), whereas progressive thinning of GCC was observed in the injured eyes (*Figure 7—figure supplement 1E*). At 3 mpi, the RGC soma density and the axonal densities at different ON sites of the injured eyes decreased dramatically compared with the contralateral eyes (*Figure 7—figure supplement 1F, G*).

## Discussion

In this study, we developed a novel TH strategy on a de novo, translatable, large animal model of CNS axonopathy. Briefly, we back-translated clinical trans-nasal endoscopic surgery into goats and non-human primates to establish a translatable TON model by crushing the pre-chiasmatic ON. This model faithfully resembles the key features of clinical TON, including specific injury site, spatiotemporal pattern of neurodegeneration, and the capability to locally modulate the injured pre-chiasmatic ON. RNA sequencing revealed that early changes were confined to the primary injury site and were enriched in inflammation, ischemia, and metabolism. Local co-delivery of hypothermia with cold-protective reagents showed significant structural and functional neuroprotection, but neither treatment was neuroprotective by itself. To ensure other research groups can faithfully replicate this endoscopy (for TH or for other ON treatments), we made a computer program to optimize surgical paths based on skull CT scans. This study potentially revolutionizes TH for CNS traumas, and provides translatable large animal models for developing local therapeutic strategies for TON and other axonal injuries.

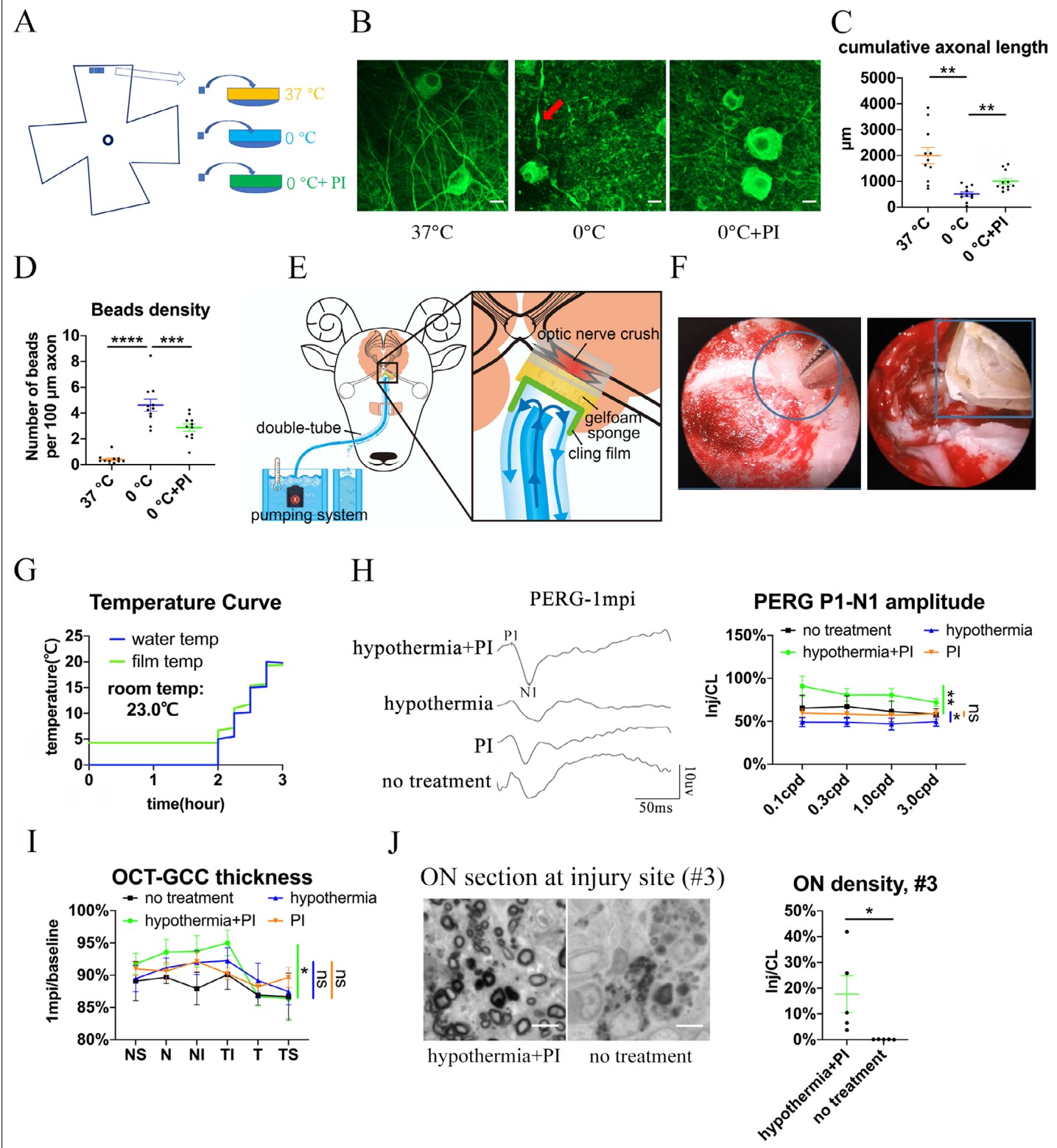

**Figure 5.** Local delivery of hypothermia combined with cold-protective reagent, but neither treatment alone, achieved significant neural protection. (**A**) Scheme for the ex vivo experiments shown in (**A**)- (**D**). (**B**) Representative confocal images of retinal explants immunostained with beta-3 tubulin (TUBB3) at 37 °C, at 0 °C, or at 0 °C with PI. A typical axonal bead was labeled by a red arrow. Scale bar: 10 µm. (**C, D**) Quantification of cumulative axonal length (**C**) and axonal beads density (**D**) at 37 °C, at 0 °C, and at 0 °C with PI. n = 11 retinal explants from 4 goats for each group. (**C**) One-way ANOVA with Dunnett's multiple comparisons test. (**D**) One-way ANOVA with Holm-Šídák's multiple comparisons test. (**E**) Scheme of local delivery of hypothermia and

*Figure 5 continued on next page*

*Figure 5 continued*

protease inhibitors (PI) at the injury site of the pre-chiasmatic ON. (**F**) Endoscopic image showing placement of a piece of sponge (shown within a circle) (left panel) and a cooling tube (shown within a rectangle) (right panel) at the injured pre-chiasmatic ON. (**G**) Temperature curves of the cooling water and the cling film during local hypothermia. (**H**) Representative PERG waveforms at spatial frequency of 0.1 cpd (left panel) and comparison of PERG P1-N1 amplitude ratios at 1 mpi in the following groups: hypothermia combined with PI, hypothermia alone, PI alone, no treatment. n = 5–6 in each group. Scheirer-Ray-Hare test with Dunn's multiple comparison. (**I**) Comparison of GCC thickness ratio of the injured eyes to the contralateral eyes at six different regions around ON head among each group at 1 mpi. n = 5–6 in each group. Two-way ANOVA and Scheirer-Ray-Hare test. (**J**) Representative images of ON semi-section (left panel) and comparison of axonal densities at the injury site in hypothermia combined with PI, and in a group with no treatment. n = 5–6 for each group. Unpaired t-test, p = 0.0398. Data were presented as mean ± s.e.m. ns: not significant, * p < 0.05, ** p < 0.01, *** p < 0.001, **** p < 0.0001. Inj: injured, CL: contralateral. The source data is in '*Figure 5—source data 1*'.

The online version of this article includes the following source data and figure supplement(s) for figure 5:

**Source data 1.** Source data for *Figure 5*.

**Figure supplement 1.** Local hypothermia combined with PI does not show significant rescue of PLR, FVEP and axonal density at distal ON segment (#4 region).

**Figure supplement 1—source data 1.** Source data for *Figure 5—figure supplement 1*.

## Current reatment for TON

TON causes severe visual loss after head trauma. Currently, no treatment for TON has been proven effective by randomized controlled trials; mostly, mere observation is recommended (*Singman et al., 2016*; *Levin et al., 1999*). In situ manipulation of the micro-environment of injured CNS axons has proven beneficial in SCI model and peripheral nerve graft transplantation studies (*Hutson and Di Giovanni, 2019*; *Aguayo et al., 1991*). However, most of the ON is hard to expose due to its deep location and crowding by neighboring tissues. Drugs delivered systemically or regionally (that is, intra-ocularly, intra-orbitally, etc.) can hardly reach the injured distal ON at sufficient and stable concentrations due to the blood brain barrier, diffusion/dilution, and impaired axonal microvasculature (*Upadhyay, 2014*; *Coles, 2004*; *Thale et al., 2002*). Our TON model allows clinically resembling local treatment *via* trans-nasal endoscopy with a large operating space, founding the basis for developing clinically translatable local therapeutic strategies for TON patients.

## Hypothermic therapy and cold-protective strategies

Systemic hypothermia may cause coagulopathy, electrolyte disturbances, myocardial ischemia, atrial fibrillation, sepsis, pneumonia, and altered drug metabolism (*Schubert, 1995*). Instead, we targetted hypothermia to the injured pre-chiasmatic ON according to early transcriptomic changes in ischemia, inflammation, and metabolism pathways (*Figure 4*). Previous study also indicated that hypothermia-induced elevated expression of cold-shock proteins may contribute to neuronal protection (*Rey-Funes et al., 2021*; *Larrayoz et al., 2016*,) especially in the mild hypothermia range (*Tong et al., 2013*; *Zhu et al., 2016*). Hypothermic treatment may directly prevent neuronal degeneration, and early treatment is necessary to alleviate further progression of neural damage after optic nerve injury (*Rey-Funes et al., 2017*) as well as after other CNS trauma (*Markgraf et al., 2001*).

Although hypothermic therapy was tested as a treatment for CNS trauma for many years, it has not shown neuroprotective effects in randomized controlled trials of TBI or SCI (*Martirosyan et al., 2017*; *Dietrich and Bramlett, 2017*). It is unclear why TH was ineffective, but cold-induced neural damage may counteract any therapeutic effects. Our recent work showed that the cold degrades neuronal microtubules, but neurons can be spared with reagents such as PI (*Ou et al., 2018*). In this study, neurons were protected when TH was combined with PI. Neither treatment alone was effective. This novel combinatory strategy can be readily applied to TON patients using established endoscopic procedures, reviving TH as a therapy for TON and other CNS traumas.

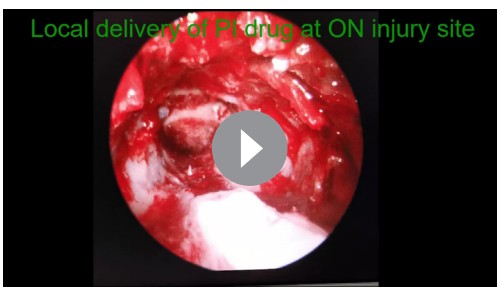

**Video 4.** Trans-nasal local delivery of hypothermia and protease inhibitors to the injured pre-chiasmatic optic nerve in a goat.
https://elifesciences.org/articles/75070/figures#video4

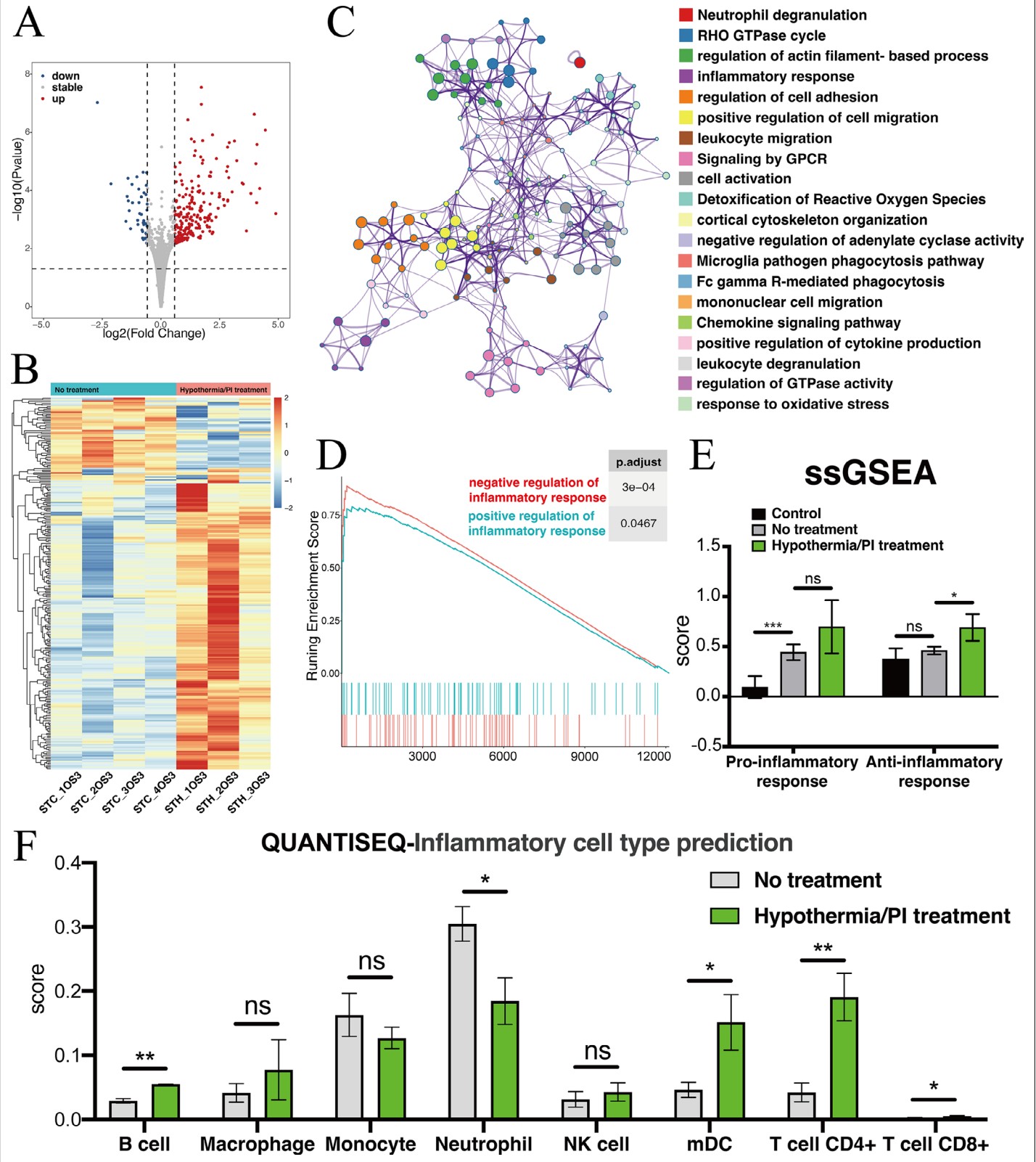

**Figure 6.** Transcriptomic changes at the injured pre-chiasmatic ON after local hypothermia/PI treatment. (**A**) Volcano plot showing the differential expression genes in the injured pre-chiasmatic ON between hypothermia/PI treatment group and no treatment group. (Threshold of p-value is 0.05, foldchange is 1.5.) (**B**) Heat map showing the inter-sample distribution of all differential genes from the difference analysis between the treatment and no-treatment groups. (**C**) Gene ontology analysis of differential genes. (**D**) GSEA enrichment profiles of two pathways: positive regulation of

*Figure 6 continued on next page*

*Figure 6 continued*

inflammatory response, negative regulation of inflammatory response. (**E**) Pro-inflammatory response (GO:0050729: positive regulation of inflammatory response) and Anti-inflammatory response (GO:0050728: negative regulation of inflammatory response) ssGSEA distribution among three groups (Control (contralateral pre-chiasmatic ON), no treatment, and hypothermia/PI treatment). (**F**) quanTIseq immune cell densities prediction analysis (mDC: Myeloid dendritic cell).

## Advantages of our TON model compared with other TON models

### Clinical translatability

Compared with the conventional retrobulbar ON crush model, our model with pre-chiasmatic ON injury resembles key clinical features of TON such as the specific injury site and the time course of neural degeneration, and allows local trans-nasal treatment using clinical endoscopic procedures and tools. As such, the pre-clinical tests of local treatment for TON succeeded in our model, and can be readily translated to TON patients. Recently, rodent models have been developed using indirect mechanisms (apply periorbital ultrasound or skull weight drop) to induce distal ON injury. Compared with direct optic nerve compression or crush, these models are likely more clinically relevant since most clinical TON cases are indirect and due to force transmission (*Evanson et al., 2018*; *Tao et al., 2017*; *Bastakis et al., 2019*). However, due to force scattering, unwanted and uncontrolled collateral damage to the eyeball, contralateral optic nerve, orbit, or skull often occurr in these models (*Evanson et al., 2018*; *Tao et al., 2017*). Additionally, the success rate of these modeling methods is not as high as direct optic nerve crush; for example, 10% mice died immediately after head weight drop (*Evanson et al., 2018*). Moreover, the extension of these modelling methods to large animal species has not been reported. Therefore, clinically translatable, local treatment of injured ONs via trans-nasal endoscopy cannot be performed in these small animal models.

Non-human primates including rhesus macaques have human-like visual systems and pathophysiological responses to CNS injury which do not exist in other mammals (*Preuss, 2004*; *Friedli et al., 2015*; *Nardone et al., 2017*.) Our work provides goat and non-human primate TON models to meet different experimental needs and budget plans for mechanistic studies or pre-clinical tests of effective treatments. Of note, the macaque model shows much more GCC thinning ( < 50%) at 3 months after injury than the goat model, likely due to much higher retinal ganglion cell density in the macaque (~1000 cells/1 mm$^2$, Figure S7) than the goat (~200 cells/1 mm$^2$) (*Zhang et al., 2020*).

### Accessibility to the injury site within a large operating space (sphenoid sinus)

In this study, histological and transcriptomic analyses found that axonal damage was initially confined to the primary injury site. The affected pathways (ischemia, inflammation, and metabolism) offer potential therapeutic targets. In our model, the chiasmatic and pre-chiasmatic ON can be exposed safely under direct visual guidance within the spacious sphenoid sinus, facilitating in situ damage, observation, and treatment of the ON with complex devices.

Traditional retrobulbar approach in rodent model allows access only to the most proximal ON within the crowded orbit. Furthermore, exposure of the retrobulbar ON is much harder in human or large animal models, and requires lateral orbitotomy, which causes transient orbital edema and retinal dysfunction (*Zhang et al., 2021*). Although access to the distal ON can be achieved by a conventional intra-cranial approach with corticectomy or a recently developed intra-orbital approach by blind injection, these approaches may cause unwanted collateral damage to the important adjacent tissues (*Mesentier-Louro et al., 2019*). In addition, the operating space is restricted in these approaches due to crowding in the cranial cavity or orbital apex.

### Facilitation eye-brain reconnection

According to our study, the axonal density of the intra-orbital ON segment remained statistically intact 1 mpi; therefore, ON regeneration can start from the pre-chiasmatic ON segment. On the contrary, the traditional retrobulbar ON crush model requires a much longer ON regeneration distance to reconnect with the brain (i.e. the distance is doubled compared with our TON model), making functional regeneration extremely challenging (*Yang et al., 2020*). In addition, our TON model allows local

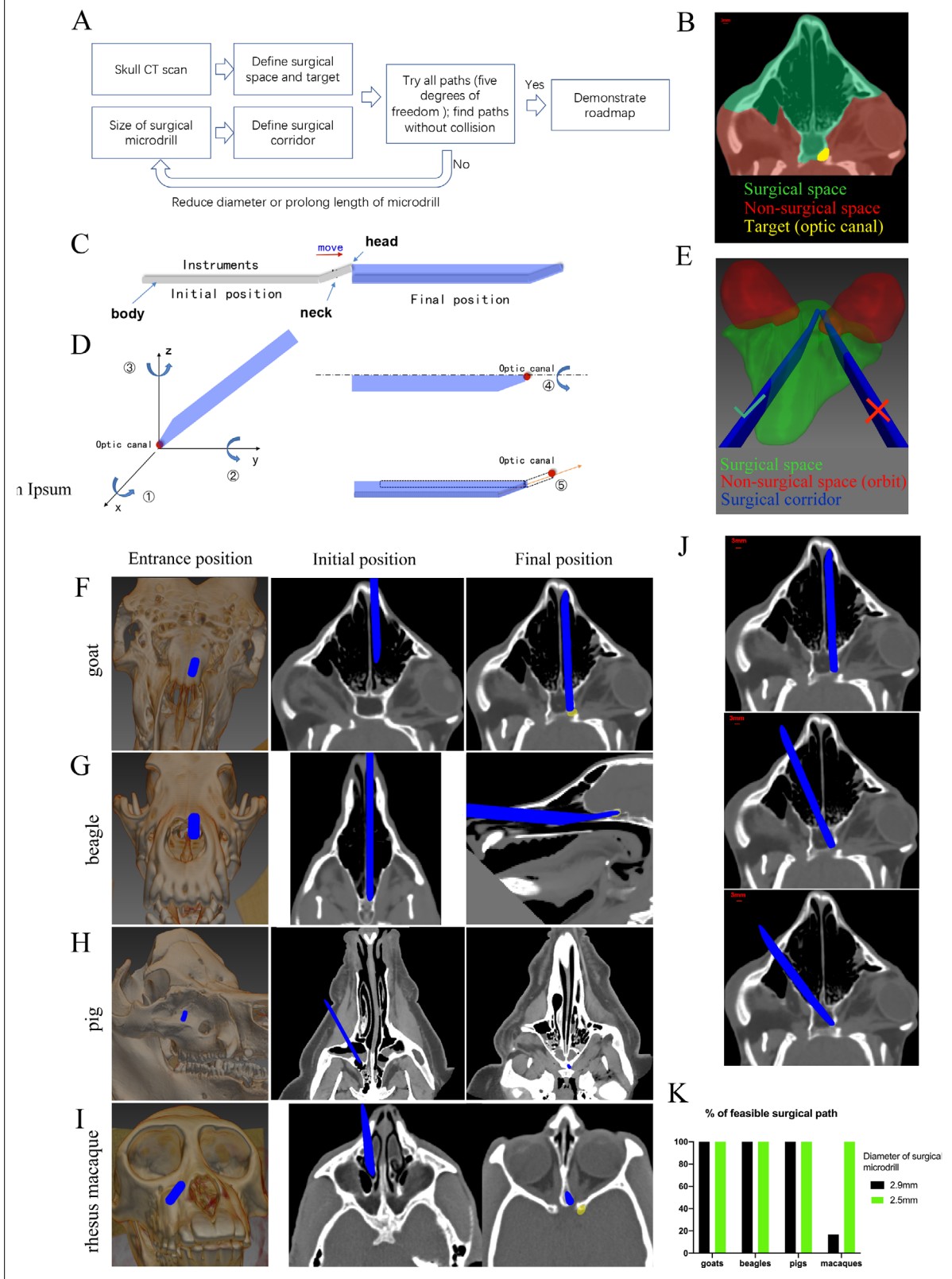

**Figure 7.** Computer program-aided optimization of trans-nasal endoscopic surgery in multiple species of large animals. (**A**) Flowchart of the computer program. (**B**) Segmentation of surgical space (in green), non-surgical space (in red) and surgical target (in yellow) in the program. (**C**) Scheme of surgical corridor (in blue) outlined by the surgical microdrill's movement along its major axis. (**D**) Demonstration of five degrees of freedom of the surgical corridor. (**E**) Demonstration of a feasible surgical corridor (on the left) and an infeasible surgical corridor colliding with orbital cavity (on the right).

*Figure 7 continued on next page*

*Figure 7 continued*

(**F–I**) Representatives of virtual surgical paths in goats, beagles, domestic pigs, and rhesus macaques, respectively. Surgical entries were shown on the surface of three-dimensional reconstructed skull (left panels) and on the horizontal CT plane (middle panels). Right panels: projection of the surgical corridor on the CT plane through the pre-chiasmatic ON (surgical target). (**J**) Representatives of multiple feasible virtual surgical paths in goats. Scale bar = 3 mm. (**K**) Quantification of percentages of feasible surgical paths in goats, beagles, domestic pigs and rhesus macaques by using different sizes of surgical microdrills (diameter = 2.5 and 2.9 mm).

The online version of this article includes the following source data and figure supplement(s) for figure 7:

**Figure supplement 1.** Pre-chiasmatic TON model in rhesus macaque (**A**) Confirmation of the pre-chiasmatic ON in rhesus macaques (left panel) using the surgical navigation system on the horizontal, coronal and sagittal CT planes (right panel).

**Figure supplement 1—source data 1.** Source data for *Figure 7—figure supplement 1*.

modulation of the microenvironment of the pre-chiasmatic and chiasmatic ON to fuel and guide ON regeneration.

## Limitations of this study

Our TON model is clinical relevant in terms of injury site, subsequent spatiotemporal pattern of retrograde axonal degeneration, and availability of trans-nasal local treatment. However, the mechanism of optic nerve injury in our model differs from that in most clinical TON cases, in which the intra-canalicular optic nerve is injured by an indirect mechanism (stretching and shearing forces), rather than by direct compressing forces. Additionally, the current local treatment did not achieve functional recovery of the eye-to-brain pathway, and its long-term therapeutic effect is unclear. In contrast, while cold treatment at the injury site has the advantages of being local, precisely controlled, and deeper hypothermia, previous articles that applied cold treatment to the whole animal also obtained positive results in vision preservation (*Rey-Funes et al., 2017*; *Rey-Funes et al., 2021*), indicating that cold treatment applied to the retina or to the whole animal may be further explored and perhaps in conjunction with local hypothermia. Another limitation of this study is a small sample size in each therapeutic group. We were aware that an analysis of sample size and power were needed for each comparison. However, due to ethical issue and limitations of housing space and other resources, we only used 3–6 goats/rhesus macaques in each group according to our previous experimental experience. Additionally, we did not repeat TH in rhesus macaques due to budget limitations and ethical concerns. In the near future, we will optimize TH with different onset times of treatment, therapeutic durations, temperature ranges, dosages, and formulae of cold-protective reagents in the goat, and then test the optimized TH strategy in the non-human primate model. Warmer temperatures in between 27°C and 33°C may achieve neuroprotection without damaging cytoskeleton, which needs to be further optimized through in vitro and in vivo studies.

## Further application of this model

In the present study, we safely exposed the chiasmatic and pre-chiasmatic ON within the spacious sphenoid sinus. This allows for several promising avenues of research. (1) Access to the ON allows for other ON disorder models, such as optic neuritis or ischemic optic neuropathy,

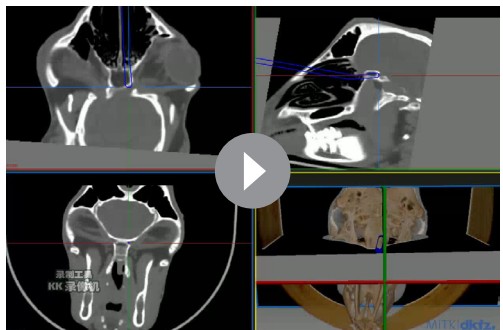

**Video 5.** Representative virtual surgical path to expose the pre-chiasmatic optic nerve in a goat.
https://elifesciences.org/articles/75070/figures#video5

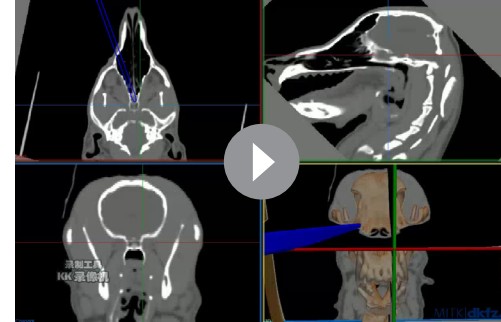

**Video 6.** Representative virtual surgical path to expose the pre-chiasmatic optic nerve in a beagle.
https://elifesciences.org/articles/75070/figures#video6

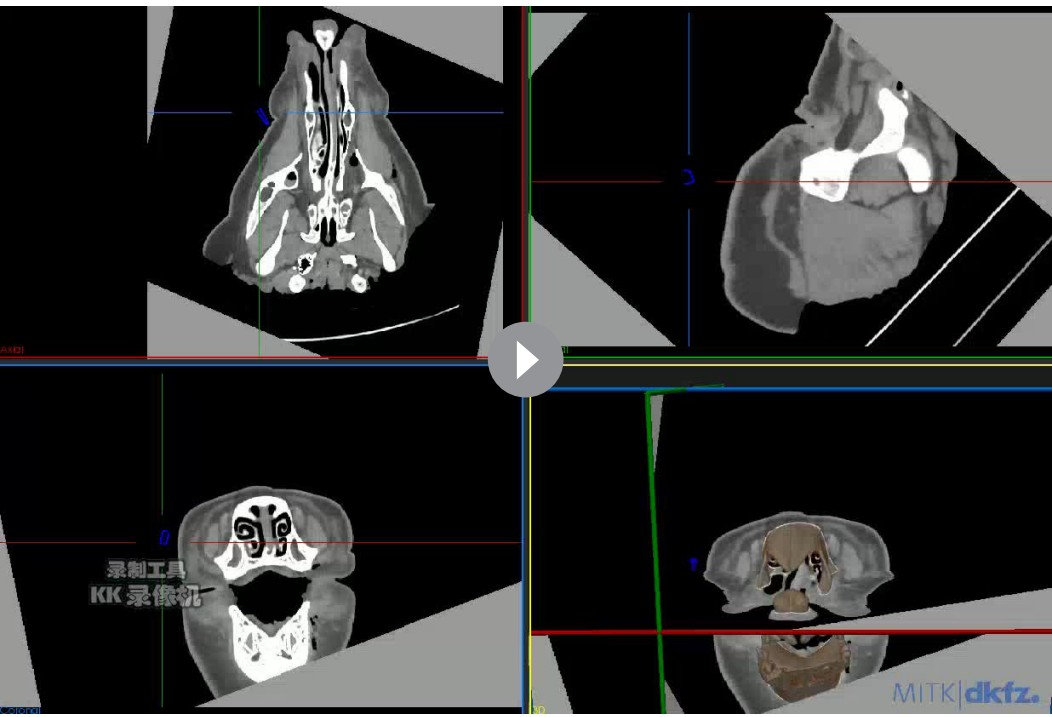

**Video 7.** Representative virtual surgical path to expose the pre-chiasmatic optic nerve in a domestic pig.
https://elifesciences.org/articles/75070/figures#video7

simply by introducing local damage. (2) Optical and electrical diagnostic tools may be placed in situ, including confocal endomicroscopy (*Belykh et al., 2019*) and endoscopic OCT. (3) In situ treatments can be tested at the pre-chiasmatic and chiasmatic ON, to prevent axonal degeneration and promote long-distance axonal regeneration by modulating the micro-environment.

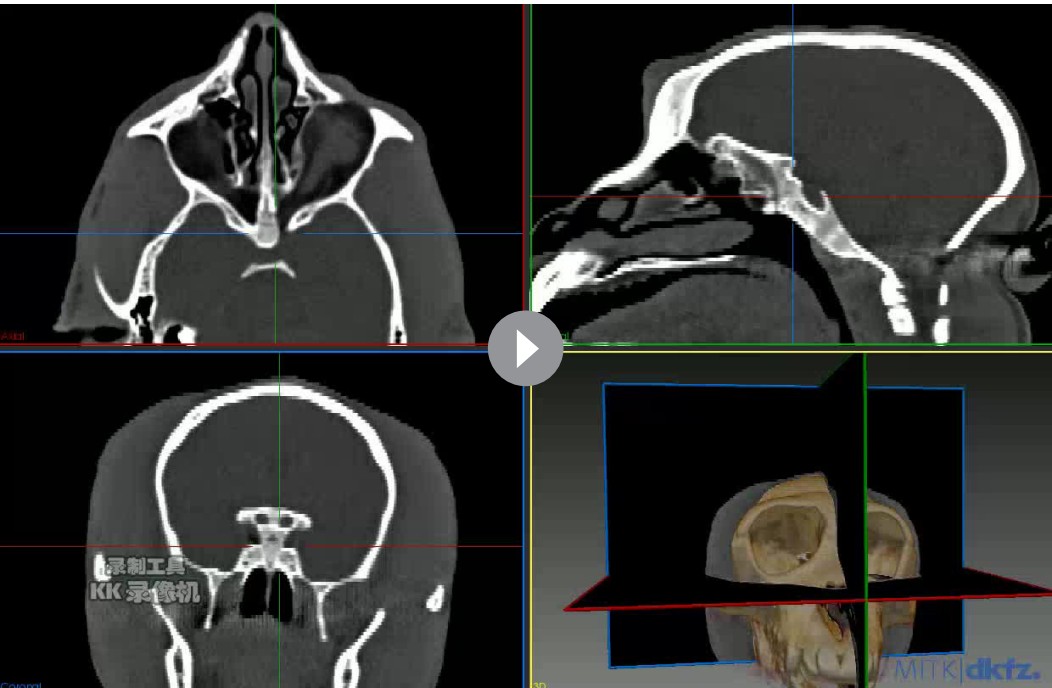

**Video 8.** Representative virtual surgical path to expose the pre-chiasmatic optic nerve in a rhesus macaque.
https://elifesciences.org/articles/75070/figures#video8

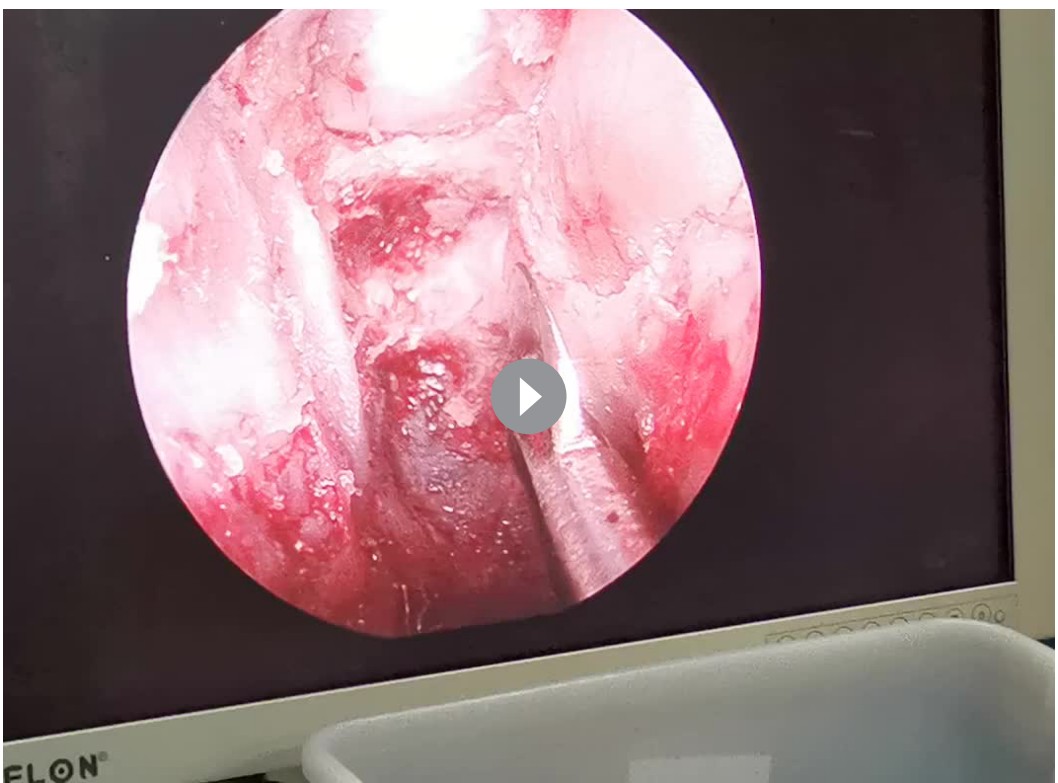

**Video 9.** Exposure of the pre-chiasmatic optic nerve via trans-nasal endoscopy in a rhesus macaque.
https://elifesciences.org/articles/75070/figures#video9

These treatments could be partnered with intra-vitreal manipulations to maximize RGC survival and ON regeneration. (4) Our work may potentially be applied in other CNS traumas such as SCI or diffused axonal injury in TBI, as well as early axonal degeneration in Alzheimer's, Parkinson's, and normal aging (*Salvadores et al., 2017*).

## Materials and methods

**Key resources table**

| Reagent type (species) or resource | Designation | Source or reference | Identifiers | Additional information |
|---|---|---|---|---|
| Biological sample (Saanen goat) | Saanen goat (4–7 months old, male) | Caimu Livestock Company (Hangzhou, China) | | |
| Biological sample (rhesus macaque) | Rhesus macaque (5–7 years old, male/female) | the Joinn Laboratory | | |
| Antibody | Anti-beta-3 tubulin (Rabbit monoclonal) | Cell Signaling Technology | Cat#5,568 S, RRID: AB_10694505 | IF(1:250) |
| Antibody | anti-Rabbit IgG (H + L) Highly Cross-Adsorbed Secondary Antibody, Alexa Fluor 488(Donkey polyclonal) | Thermo Fisher Scentific | Cat# A-21206, RRID:AB_2535792 | IF(1:200) |
| Antibody | Anti-RBPMS(Guinea pig polyclonal) | custom-made by ProSci | PMID:31090540 | IF(1:4000) |
| Antibody | Cy3-conjugated, anti-guinea pig(Donkey polyclonal) | Jackson Immuno Research | Cat#706-165-148, RRID: AB_2340460 | IF(1:200) |
| Chemical compound, drug | Triton X-100 | Sigma | Cat#T8787 | dilution (1:200) |

*Continued on next page*

*Continued*

| Reagent type (species) or resource | Designation | Source or reference | Identifiers | Additional information |
|---|---|---|---|---|
| Chemical compound, drug | PPD | Macklin | Cat#P816017 | |
| Chemical compound, drug | Methanol | Macklin | Cat#P813895 | |
| Chemical compound, drug | Isopropanol | Macklin | Cat#I811925 | |
| Chemical compound, drug | Uranyl acetate | Lanzhou 404 Factory | Cat#8,901 | |
| Chemical compound, drug | TRIzol Reagent | Life Technologies | Cat#15596–026 | |
| Chemical compound, drug | 5% povidone-iodine solution | Zhejiang Apeloa Inc. | | |
| Chemical compound, drug | Protease inhibitor | Miilipore | Cat#539,134 | |
| Chemical compound, drug | Hibernate A medium | BrainBits | Cat# 2283833 | |
| Chemical compound, drug | 4% PFA | Servicebio | Cat#G1101 | |
| Chemical compound, drug | 4% glutaraldehyde | Phygene | Cat#PH1021 | |
| Chemical compound, drug | Ethanol | Sinopharm Chemical Reagent CO.,Ltd | Cat#100092183 | |
| Chemical compound, drug | EMbed 812 | SPI | Cat#90529-77-4 | |
| Chemical compound, drug | Propanone | Sinopharm Chemical Reagent CO.,Ltd | Cat#10000418 | |
| Chemical compound, drug | Osmium tetroxide | Ted Pella Inc | | |
| Chemical compound, drug | Zoletil50 | Virbac | | 4–8 mg/kg, IM |
| Chemical compound, drug | Xylazine | Huamu Animal Health Products, China | | 4 mg/kg, intramuscular (IM) |
| Chemical compound, drug | Idzoxan | Huamu Animal Health Products, China | | 1.5 mg/kg, IM |
| Chemical compound, drug | Propofol | Xian Lipont Enterprise Union Management, China | | 5 mg/kg, intravenous (IV) |
| Chemical compound, drug | Isoflurane | RWD Life Science, China | | 2.5%–3.5%, by mechanical ventilator |
| Chemical compound, drug | Atropine | Guangdong Jieyang Longyang Animal Pharmaceutical, China | | 0.05 mg/kg, IV |
| Chemical compound, drug | Esmolol hydrochloride | Qilu Pharmaceutical, China | | 0.2 g (2 ml) in 48 ml saline, IV drip infusion |
| Chemical compound, drug | Hemocoagulase Atrox | Penglainuokang Pharmaceutical, China | | 1 unit (1 shot), IM |
| Chemical compound, drug | Dexamethasone | Kelong Veterinary Medicine, China | | 5 ml (5 mg), IV |
| Chemical compound, drug | Ceftiofur sodium | Jiangxi Huatu, China | | 20 mg/kg, IM |

*Continued on next page*

*Continued*

| Reagent type (species) or resource | Designation | Source or reference | Identifiers | Additional information |
|---|---|---|---|---|
| Chemical compound, drug | Gentamicin | Shanxi Ruicheng Kelong Co., Ltd, China | | 8 mg/kg, IM |
| Chemical compound, drug | Potassium chloride | Bei 'an Feilong Animal Medicine, China | | 40 mL, IV |
| Commercial assay or kit | NEBNext Ultra RNA Library Prep Kit | New England BioLabs | Cat#E7530 | |
| Commercial assay or kit | NEBNext Poly (A) mRNA Magnetic Isolation Module | New England BioLabs | Cat# E7490 | |
| Software, algorithm | GraphPad | GraphPad | RRID:SCR_002798 | 9.0 |
| Software, algorithm | R Project for Statistical Computing | R Project | http://www.r-project.org/ | v4.1.2 |
| Software, algorithm | Image J | Image J | RRID:SCR_003070 | v1.53a |
| Software, algorithm | MITK | MITK | http://mitk.org/wiki/Downloads | v2018.04.2 |
| Software, algorithm | Computer-assisted surgical path planning program | This paper | https://github.com/LujieZhang/Preoperative-planning | |
| Software, algorithm | HISAT2 | Github | http://daehwankimlab.github.io/hisat2/ | v2.0.4 |
| Software, algorithm | StringTie | The Center for Computational Biology at Johns Hopkins University | http://ccb.jhu.edu/software/stringtie/ | v1.3.1 |
| Software, algorithm | NetworkAnalyst | Xia Lab | https://www.networkanalyst.ca | v3.0 |
| Software, algorithm | Bcl2fastq | illumina | https://support.illumina.com/sequencing/sequencing_software/bcl2fastq-conversion-software.html | v2.19.1 |
| Software, algorithm | FastQC | Babraham Bioinformatics | http://www.bioinformatics.babraham.ac.uk/projects/fastqc | |
| Software, algorithm | Metascape | Metascape | http://metascape.org/gp/#/main/step1 | |
| Software, algorithm | Cytoscape | Cytoscape | https://cytoscape.org | v3.8.2 |
| Other | NEBNext Multiplex Oligos | New England BioLabs | Cat#750 | Construct the cDNA library |
| Other | Goat serum | Beyotime Institute of Biotechnology | Cat# C0265 | Dissolvant of PI |

## Study Design

This study sought to establish a de novo, clinically-relevant, large animal TON framework, and to develop a novel local TH combined with cold protection. Due to ethical issue and limitations of housing space and other resources, we only used 3–6 goats/rhesus macaques in each group according to our previous experimental experience. At least THREE animals were used per test. For in vivo tests, histology, and transcriptomic studies, sampling and experimental replicates are detailed in the Methods. For ex vivo studies, retinal explants were randomly assigned to three experimental groups (0 °C, 0 °C with PI, 37 °C). For in vivo studies, goats from the same company, of the same gender, and of similar ages and weights, were allocated to the groups of injury, hypothermia combined with PI, hypothermia alone, and PI alone without randomization. All the surgeries were performed by one surgeon, and the investigators collecting and analyzing the data were blinded to the grouping. No data outliers were excluded.

## Saanen goats and rhesus macaques

Experiments were conducted following the Association for Research in Vision and Ophthalmology (ARVO) Statement for the Use of Animals in Ophthalmic and Vision Research guidelines. All protocols were approved by the Institutional Animal Care and Use Committee in the Wenzhou Medical University

(Wenzhou, China, ID number: wydw2020-0789) and the Joinn Laboratory (Suzhou, China, ID number: P19-S445-PD). Four- to 7 month-old male Saanen goats weighing 19–22 kg were purchased from the Caimu Livestock Company (Hangzhou, China) and housed in the animal facility of the Wenzhou Medical University. The rhesus macaques aged from 5 to 7 years weighing 5–7 kg were housed in the animal facility of the Joinn Laboratory. All animals were housed in an air-conditioned room (21°C ± 2°C) under a normal light/dark cycle (12/12 hr) with food ad libitum. Goats were euthanized with 10% potassium chloride (40 ml, intravenously) after general anesthesia using xylazine (3 mg/kg, IM). Rhesus macaques were euthanized by exsanguination after general anesthesia using Zoletil50 (4–8 mg/kg, IM, tiletamine/zolazepam).

## Trans-nasal endoscopic procedure to expose the pre-chiasmatic on in goats and rhesus macaques

Goats were anesthetized with xylazine, and then intubated with a 6.0 mm tracheal tube (Henan Tuoren Medical Device Co., Ltd, China). Anesthesia was maintained with 3% isoflurane in oxygen and air (1:1) at a rate of 2 L/min by using a mechanical ventilator. Then atropine, ceftiofur sodium, gentamicin, dexamethasone, esmolol hydrochloride, and hemocoagulase atrox were injected (detailed information of each drug is listed in the 'Key Resources Table').

After skin preparation with povidone-iodine solution (20 ml, 5%, Zhejiang Apeloa Inc, China), a vertical double-T-shaped incision on the nose was made through the skin and periosteum. Following blunt dissection, the underlying nasal bone was removed to access the nasal cavity. Under trans-nasal endoscopy (Delong, HD380B), the middle turbinate and posterior olfactory nerve filaments were partially removed by an endoscopic microdebrider (Medtronic, 1884004) powered by the Integrated Power Console system (Medtronic, Integrated Power Console ENT Surgery, 1898001) to expose the ipsilateral anterior wall of the sphenoid bone. To increase surgical operating space, the nasal septum and part of the contralateral olfactory nerve filaments were also resected to fully expose the anterior bony wall of the sphenoid bone (*Figure 1E*, **left panel**). To access the pre-chiasmatic ON within the sphenoid bone, an endoscopic microdrill (Medtronic, diamond microdrill, 1882969) was employed to drill the overlying cortical bone. The posterior myeloid tissue within the sphenoid bone body (*Figure 1E*, **right panel**) was aspirated and drilled to create an artificial sphenoid sinus, within which the chiasmatic and pre-chiasmatic ON lay posteriorly (*Figure 1F*). There are two ways to identify the optic canal (part of the pre-chiasmatic ON) in goats during the endoscopic surgery, either by exposing the anterior bony wall of the optic chiasm and then pinpointing the optic canal at the junction of the optic chiasm and the medial orbital wall (*Figure 1G*) or by using a surgical navigation system.

In the sham surgery group, the ON was exposed by drilling off its anterior bony wall without ON crush (*Figure 1G*). In the ON crush injury group: the ON was crushed by using the blunt blade (3 mm long, 0.5 mm wide) of a periosteal elevator (HBB010, Shanghai Medical Instruments, China) for 5 s (*Figure 2A and B*).

Following ON injury, the nasal cavity was irrigated consecutively with povidone-iodine and saline, and the artificial sphenoid sinus was filled with an absorbable gelatin sponge (Xiangen Medical Technology, China). After closure of the nasal periosteum and the skin with 3–0 sutures, anesthesia recovery was conducted.

The trans-nasal endoscopic procedure to expose the optic canal in rhesus macaque was similar to that in goats and used the same surgical equipment. Our self-designed surgical path planning program helped screen for suitable macaques and determine the surgical path based on their pre-surgical cranial CT scans (*Video 9*). The rhesus macaque was anesthetized by isoflurane (1.5–2%) after induction by Zoletil50 (4–8 mg/kg IM, Virbac, France) in the supine position. The exposed pre-chiasmatic ON was confirmed by a surgical navigation system (NDI Polaris Vicra).

## Computer-assisted semi-automated surgical path planning in trans-nasal endoscopic surgery based on CT scans

To facilitate endoscopic exposure and expand this surgical approach to other large animal species, a computer-assisted surgical path planning program was designed based on pre-surgical cranial CT scans. The flowchart of the program is shown in *Figure 7A*. The major steps were as follows:

1. Virtual reconstruction of surgical space based on cranial CT scans. Cranial CT scans with slice thicknesses of 0.65 mm in goats and 1 mm in beagles and macaques were obtained before

surgery (pigs and goats, GE Optina660, General Electric Company, Boston, USA; beagles and macaques, Philip Brilliance iCT, Royal Philips Company, Amsterdam, Holland). The surgical space was manually segmented in the MITK software (MITK, v2018.04.2, detailed code is available in http://mitk.org/wiki/Downloads). The anatomic boundaries of the surgical space were as follows: the nasal and maxillary bone anteriorly, the anterior skull base superiorly, the palate inferiorly, the medial orbital walls laterally, and the posterior wall of the sphenoid body posteriorly (*Figure 7B*, **in green**). The anterior wall of the left optic canal was marked as the surgical target (*Figure 7B*, **in yellow**) that needed to be approached by the surgical microdrill.

2. Virtual reconstruction of the surgical microdrill and the surgical corridor (formed by the movement of the surgical microdrill along its major axis) (*Figure 7C*).

3. Calculation and optimization of the surgical path. There were five degrees of freedom for the movement of the surgical corridor when its tip met the optic canal (*Figure 7D*). Using a comprehensive search in the program, all possible surgical paths in the three-dimensional virtual space were tested and screened for the most feasible paths in which all the pixels within the surgical corridor were either in the confined surgical space or in air (space outside of the animal's skull) (*Figure 7E*).

4. Optimization of microdrill size. If no feasible surgical path was identified, the computer automatically decreased the diameter of the surgical microdrill by 0.5 mm, and then repeated steps (1) - (3) to screen for feasible paths.

The code of the computer program is available online (https://github.com/LujieZhang/Preoperative-planning; *Zhang, 2022*; copy archived at swh:1:rev:f718fe67d388d3bbd6ef382499b8485af4aaa06a).

## Evaluation of microtubule cold-stability in the retinal explants of goats

Goat eyecups were prepared within 15 min after euthanization with 10% potassium chloride (40 ml, intravenously) in conjunction with general anesthesia using xylazine. Three neighboring retinal pieces, 2 mm in diameter, were taken from the peripheral retina, and randomly assigned to three experimental groups (0 °C, 0 °C with PI (cold-protective reagent), 37 °C) (*Figure 5A*). The retinal explants were pre-treated in hibernate A medium (BrainBits, 2283833, USA) with or without PI (1:500, EMD Millipore Company, 539134, Germany) for 30 min at room temperature, then were kept in a 37 °C incubator or in a 0 °C ice-water mixture for 6 hr. After fixation in 4% PFA for 30 min, the retinal explants were immuno-stained against microtubule with beta-3 tubulin (TUBB3) antibody (1:250, from rabbit, Cell Signaling Technology, 5,568 S, USA), washed in 0.5% Triton in PBS for 5 times, and then were immune-stained with Alexa Fluor 488 conjugated anti-rabbit secondary antibody (1:200, from donkey, Thermo Fisher Scentific, A-21206, USA). Confocal images were taken under confocal microscopy (Cell Observer SD, ZEISS, Germany) with 63 X oil objective lens. Accumulative axonal length of TUBB3-positive optic nerve was analyzed using computer-assisted software ('Simple Neurite Tracer' plugin, ImageJ). Axonal beads along the TUBB3-positive optic nerve were counted manually and bead density was calculated as the number of beads per 100 μm TUBB3-positive optic nerve. For each retinal explant, three regions (108.36 μm x 108.36 μm, 512 × 512 pixels) were imaged, and the accumulative axonal length and axonal bead densities in these three regions were averaged to yield one readout. Retinal pieces with many optic nerve bundles were excluded to facilitate precise quantification. Eleven retinal explants from 4 male goats were used in each experimental group. After testing for normality, a repeated measures one-way ANOVA was applied with Dunnett's multiple comparisons test.

## Trans-nasal local hypothermic therapy

We developed a trans-nasal local cooling system. Cooling water was pumped into a nasal irrigation tube by a rotary pump (Beijing Zhongshidichuang Technology Development Co., Ltd, China). An irrigation and drainage system consisted of a double-tube cooling device including a pair of concentric silicon tubes. The inner silicon tube (irrigation tube, 3 mm in diameter, Zhejiang Kangkang Medical Equipment Co., Ltd, China) was connected to the pump to receive the cool water. The outer silicon tube (drainage tube, 7 mm in diameter, Taizhou Biling Hardware Products Co., Ltd., China) was covered with cling film (Shanghai Runwen Packaging Materials Co., Ltd., China) to drain the water out of the nasal cavity (*Figure 5E*). A gelfoam sponge (Xiang'en Jiangxi Medical Technology Development Co., Ltd., China) filled with goat serum (Beyotime Institute of Biotechnology, C0265, China) with or without 1:200 diluted PI (EMD Millipore Company, 539134, Germany) was placed

between the injured ON and the cling film (*Figure 5E and F*). To test the effect of hypothermia and PI (cold-protective reagents) on injured ONs, goats were assigned to the following three experimental groups: (1) local hypothermia with PI, (2) local hypothermia without PI, and (3) PI without local hypothermia.

Immediately after the ON crush injury, the sponge was placed at the injury site and then the local cooling tubes were inserted over the gelfoam sponge (*Video 4*). Local hypothermia started within 20 min after optic nerve crush. The iced water mixture (0 °C as measured by a digital thermometer; ZS-T, Beijing Zhongshidichuang Technology Development Co., Ltd., China) was pumped into the double-tube device to cool the outer surface of the cling film to 4 °C (measured by the same digital thermometer). After 2 hr of continuous cooling, the iced water mixture was replaced consecutively with water at 5, 10, 15, and 20 °C water for 15 min each to achieve slow rewarming (*Figure 5G*). Then the goat was awakened from general anesthesia. Room temperature was maintained at 23 °C.

To explore transcriptomic changes after local treatment with deep hypothermia and PI, longer (6 hours) local hypothermia was applied. Previous studies in SCI shown longer hypothermia may result in better therapeutic effect (*Martirosyan et al., 2017*).

## RNA-sequencing

Fresh samples from the goat's retina, retrobulbar, and pre-chiasmatic ON in injury and contralateral eyes were harvested and frozen in liquid nitrogen within 30 min after euthanization at 1 dpi (*Figure 4A*). RNA extraction and sequencing was completed in a bio-company (Biomarker Technologies, China). Total RNA of each sample was extracted following the instruction manual of the TRIzol Reagent (Life Technologies, CA, USA). Agilent 2,100 Bioanalyzer (Agilent Technologies, Inc, Santa Clara, CA, USA) was used to test the RNA integrity and concentration. The mRNA was isolated by NEBNext Poly (A) mRNA Magnetic Isolation Module (NEB, E7490), then fragmented into approximately 200 nt RNA inserts. The cDNA library was constructed according to the NEBNext Ultra RNA Library Prep Kit for Illumina (NEB, E7530) and NEBNext Multiplex Oligos for Illumina (NEB, E750), and was sequenced using an Illumina HiSeq sequencing platform. The reads were mapped to ARS1 (*Capra hicus*) genome (GenBank assembly accession: GCA_001704415.1) using HISAT2 and StringTie software. Gene expression levels were estimated using Log2 CPM transformation values on NetworkAnalyst 3.0 website. The raw base call (.bcl) files were converted to demultiplexed FASTQ files with Bcl2fastq v2.19.1 for data analysis. The RNA-sequencing and analysis were repeated in 4 goats in no treatmen group and 3 goats in hypothermia/PI group. The RNA-seq data has been uploaded online (https://www.ncbi.nlm.nih.gov/geo/query/acc.cgi?&acc=GSE182164 and https://www.ncbi.nlm.nih.gov/geo/query/acc.cgi?acc=GSE197123).

## RNA-Seq analysis

Raw reads were assessed using the FastQC toolset (http://www.bioinformatics.babraham.ac.uk/projects/fastqc). The statistics on the raw reads and alignments are shown in sheet 'Mapping Statistics', Data S3. R Subread package was used to quantify gene expression. Adjusted p values from differential expression tests were calculated using the Benjamini-Hochberg procedure for multiple hypothesis testing. To further explore the expression differences between samples, DESeq2 R package was used for differential analysis. Genes were sorted by the shrunk fold change as computed by the 'lfcShrink' function to obtain a more accurate differential expression fold change of genes with fewer biological replicates. Gene ontology (GO) analysis was performed based on hypergeometric test using clusterProfiler R package. Gene list analysis was performed by Metascape website (http://metascape.org/gp/#/main/step1) using default setting. GSEA analysis was completed using clusterProfiler R package. ssGSEA scores were calculated using the GSVA Bioconductor package.

To obtain the downstream protein-protein interaction map, the differentially expressed genes were analyzed in the STRING database (https://string-db.org/). Genes that interact more with other genes at the protein level were analyzed in the Cytoscape software (version 3.8.2). The genes with high connectivity were named as hub genes. The immune cell infiltration status was acquired based on quanTIseq, wihich performs a supervised deconvolution to quantify the fractions of immune cell types from bulk RNA-sequencing data.

### Flash visual evoked potential (FVEP)

Visual evoked potential (FVEP) is the cortical potential in response to visual stimulus, reflecting the function of retino-geniculate pathway (*Odom et al., 2010*). FVEP recording in goats has been previously reported in detail (*Zhang et al., 2020*). Briefly, following general anesthesia and electrode placement, the animal was adapted to the testing environment for 5 min. Then the FVEP was recorded at intensities of 0.025 and 0.25 cd·s/m$^2$ consecutively (GT-2008V-Ⅲ, GOTEC Co., Ltd, China). To reduce variation, we used the FVEP amplitude ratio of the surgical eye to the contralateral eye to quantify the FVEP changes in the surgical eyes.

The pattern visual evoked potential (PVEP) test in rhesus macaques was described previously (*Zhang et al., 2020*). The PVEP, which is more reproducible than FVEP in human (*Odom et al., 2010*), can be elicited in rhesus macaques according to our previous study (*Zhang et al., 2020*). Therefore, we replaced the FVEP test with the PVEP test in rhesus macaques.

### Pattern electroretinogram (PERG) recordings

PERG is the retinal electrical signal activated by patterned visual stimuli, the amplitude of which indicates the functional integrity of RGC (*Porciatti, 2015*). PERG recording in goats and rhesus macaques has been previously reported (*Zhang et al., 2020*). Briefly, the goat was anesthetized with xylazine. After electrode placement, PERG signals were elicited by contrast-reversal black-white checkerboards (temporal frequency, 2.4 Hz) at spatial frequencies of 0.1, 0.3, 1.0, 3.0 per degree consecutively (GT-2008V-Ⅲ, GOTEC Co., Ltd, China). To reduce variation, we report a ratio of the PERG amplitude of the surgical eye to the contralateral eye.

### Pupillary light response (PLR) test

The PLR test was used to evaluate the integrity of the ocular afferent (ON) and the efferent (oculomotor nerve) pathways (*Kerrison et al., 2001*). Direct PLR (dPLR) is the PLR in the ipsilateral eye while indirect PLR (iPLR) occurs in the contralateral eye. This method has been previously described (*Zhang et al., 2020*). Briefly, the goat was dark-adapted for 5 min after anesthesia with xylazine. Two infrared cameras were positioned at 5 cm from each eye and focused on the pupillary plane to film the PLR. Pupillary constriction was elicited by white light stimulus (230 lx) for two seconds. The PLR results were scored manually by the same person as follows: Grade 2 (score = 2) = normal PLR; Grade 1 (score = 1) = delayed or weak PLR; Grade 0 (score = 0) = no PLR.

### Spectral-domain optical coherence tomography (SD-OCT) imaging

Retinal OCT imaging was used to measure the thickness of the retinal ganglion cell complex including the RNFL, RGC and IPL layers, which represent the RGC axons, somas and dendrites, respectively. This method has been previously described (*Zhang et al., 2020*). Briefly, after anesthesia, the OCT images were taken with peripapillary circular scan pattern (Heidelberg Spectralis OCT system, Germany). The thickness of the GCC was measured manually using the Heidelberg software.

### Immunohistochemical (IHC) staining of retinal ganglion cells (RGCs)

The IHC staining for RGCs was performed as described in our previous study (*Zhang et al., 2020*). Briefly, the eyecup was made and fixed in 4% paraformaldehyde in phosphate-buffered saline (PBS) for 24 hr at 4 °C. For each retinal quadrant, three circular retinal pieces with a radius of 1 mm were dissected out using a corneal trephine blade (Zhonglindongsheng Medical Instrument, Jiangsu, China) at the central, mid-peripheral and peripheral retina (1/4, 1/2 and 3/4 of the retinal radius away from the ON head, respectively). For each eye, the RGC densities of 12 retinal pieces were averaged to yield one readout. Retinal samples were blocked in 5% goat serum (C0265, Beyotime Institute of Biotechnology, China) in 0.5% PBST overnight, and then incubated in a 1:4000 diluted primary antibody (anti-RBPMS from guinea pig, ProSci, California, custom-made) in 0.5% PBST for 24 hr at room temperature. After being washed in PBS, the retinal pieces were incubated in a 1:200 diluted secondary antibody in PBS (Cy3-conjugated, anti-guinea pig from donkey, Jackson Immuno Research, West Grove, Pennsylvania) for 12 hr at room temperature. The retinal pieces were then washed before being mounted on slides. Confocal images were taken using a Zeiss LSM710 system (Carl Zeiss Meditec, Sartrouville, Germany) under a 20 X objective lens. RBPMS-positive RGCs were counted manually using ImageJ software (NIH, Bethesda, MD, USA).

## Quantification of surviving axons within a semi-thin section

Dissection of the ON was completed within 30 min after euthanization. The ON was cross-sectioned into 1–2 mm-thick discs at the retrobulbar (2 mm behind the eyeball, named as #1), mid-orbital (#2), intra-canalicular (lesion site, #3) and pre-chiasmatic (post-injury site, #4) regions. The ON sections were fixed in 2% glutaraldehyde and 2% PFA in PBS for 24 hr at 4 °C. After being washed in 0.1 M PB, the samples were incubated in 1% osmium tetroxide in 0.1 M PBS for 1 hr and then incubated in 2% uranyl acetate in double distilled water for 1.5 hr at 37 °C. The samples were then dehydrated through a series of graded ethanol (50–100% in H2O) for 10 min at each concentration. The samples were then embedded in 50% EMbed 812/50% propanone for 1 hr, followed by a 4:1 ratio of EMbed 812/propanone overnight at 37 °C. The next day, the samples were transferred to 100% EMbed 812 for 1 hr at 45 °C and embedded in a mold filled with 100% EMbed 812 at 45 °C for 3 hr and then at 65 °C for 2 days. Semi-thin sections (2 μm) were cut using an ultramicrotome (LKB-2088, LKB, Bromma, Sweden), stained with 1% para-phenylenediamine (PPD) in methanol: isopropanol (1:1) for 35 min and then rinsed three times with methanol:isopropanol (1:1). Myelin and unhealthy axonal cytoplasm were stained with PPD. More than five separate regions (125.22 μm x 94.26 μm) of each section were imaged at 70 X magnification using a Leica DM4B epifluorescence microscope. The surviving axons were semi-automatically counted to obtain the average surviving axonal density for each section. Since the axonal densities amongst different ON sites of the contralateral eyes were statistically the same at 1 and 3 mpi after ON crush, we used the axonal density at region #3 to represent the axonal density in the contralateral eye.

## Materials and correspondence

Correspondence and material requests should be addressed to Yikui Zhang, 86–13705770161, zhang.yikui@wmu.edu.cn.

## Data and materials availability

Computer program download site: https://github.com/LujieZhang/Preoperative-planning; *Zhang, 2022*; copy archived at swh:1:rev:f718fe67d388d3bbd6ef382499b8485af4aaa06a).

The processed gene expression data in this paper have been deposited into the NCBI GEO database: GSE182164,, GSE197123. RNA-seq data download site: https://www.ncbi.nlm.nih.gov/geo/query/acc.cgi?&acc=GSE182164; https://www.ncbi.nlm.nih.gov/geo/query/acc.cgi?acc=GSE197123.

## Statistical analyses

All data were analyzed using GraphPad (9.0) software or R Project for Statistical Computing (4.1.2). Normality tests were used to analyze the distributions of all data sets. Residual analysis was used to check the assumption of normal distribution of the raw data before two-way ANOVA analysis, because a recent study showed that normality test on the residuals is better for checking ANOVA assumptions than normality test on the raw data (*Kozak and Piepho, 2018*). To compare two groups of data, Student's t-test or nonparametric test was used. To compare multiple groups, a one-way ANOVA (with Dunnett's multiple-comparisons test) or nonparametric Kruskal-Wallis test (with Dunn's multiple-comparisons test) was used. Two-way ANOVA (with multiple comparisons) was used to analyze OCT, PERG, FVEP, PVEP and axonal density data. Asterisks (*) represent statistically significant differences (* $p < 0.05$, ** $p < 0.01$, *** $p < 0.001$, **** $p < 0.0001$). Data are presented as mean ± s.e.m. Statistical details are included in the supplementary file named 'statistics.xlsx'.

## Acknowledgements

We thank Dr. Haohua Qian for critically reading this manuscript. We thank Dr. Michael W Country for editing this manuscript. We also appreciate Professor Kaihui Nan, Yu Xia, Yuanfei Ji, Mingna Xu, Qiqi Xie, Weijie Liu, Zhaoqi Pan, Xiaohui Jiang, Yao Zhou, Mengting Jin, Haochen Jin, Ke Li, Junbo Chen for taking part in experimental conduction.

## Additional information

### Funding

| Funder | Grant reference number | Author |
| --- | --- | --- |
| National Key Research and Development Program of China | 2021YFA1101200 | Wencan Wu |
| National Key Research and Development Program of China | 2016YFC1101200 | Wencan Wu |
| National Natural Science Foundation of China | 81770926 | Wencan Wu |
| National Natural Science Foundation of China | 81800842 | Yikui Zhang |
| Key R&D Program of Zhejiang Province | 2018C03G2090634 | Wencan Wu |
| Key R&D Program of Zhejiang Province | 2021C03065 | Wencan Wu |
| Key R&D Program of Wenzhou Eye Hospital | YNZD1201902 | Wencan Wu |
| National Key Research and Development Program of China | 2019YFC0119300 | Jian Yang |
| National Eye Institute | EY023295 | Yang Hu |
| National Eye Institute | NEI Intramural Research Program | Wei Li |

The funders had no role in study design, data collection and interpretation, or the decision to submit the work for publication.

### Author contributions

Yikui Zhang, Software, Conceptualization, Funding acquisition, Investigation, Methodology, Software, Supervision, Validation, Visualization, Writing - original draft, Writing - review and editing; Mengyun Li, Shengjian Lu, Funding acquisition, Investigation, Visualization; Bo Yu, Senmiao Zhu, Zhonghao Yu, Tian Xia, Haoliang Huang, WenHao Jiang, Si Zhang, Lanfang Sun, Qian Ye, Jiaying Sun, Hui Zhu, Pingping Huang, Huifeng Hong, Funding acquisition, Investigation; Lujie Zhang, Wenjie Li, Danni Ai, Jingfan Fan, Wentao Li, Hong Song, Funding acquisition, Investigation, Software; Shuaishuai Yu, Lei Xu, Xiwen Chen, Funding acquisition; Tongke Chen, Investigation; Meng Zhou, Software; Jingxing Ou, Funding acquisition, Investigation, Writing - review and editing; Jian Yang, Software, Conceptualization; Wei Li, Software, Funding acquisition, Investigation, Writing - review and editing; Yang Hu, Software, Writing - review and editing; Wencan Wu, Software, Conceptualization, Methodology, Supervision, Writing - review and editing

### Author ORCIDs

Yikui Zhang ⓘ http://orcid.org/0000-0001-9060-602X
Mengyun Li ⓘ http://orcid.org/0000-0001-8422-9061
Wei Li ⓘ http://orcid.org/0000-0002-2897-649X
Yang Hu ⓘ http://orcid.org/0000-0002-7980-1649

### Ethics

Experiments were conducted following the Association for Research in Vision and Ophthalmology (ARVO) Statement for the Use of Animals in Ophthalmic and Vision Research guidelines.All protocols were approved by the Institutional Animal Care and Use Committee in the Wenzhou Medical University (Wenzhou, China, ID number: wydw2020-0789) and the Joinn Laboratory (Suzhou, China, ID number: P19-S445-PD).

Decision letter and Author response
Decision letter https://doi.org/10.7554/eLife.75070.sa1
Author response https://doi.org/10.7554/eLife.75070.sa2

## Additional files

### Supplementary files
• Transparent reporting form

### Data availability
Computer program download site: https://github.com/LujieZhang/Preoperative-planning, (copy archived at swh:1:rev:f718fe67d388d3bbd6ef382499b8485af4aaa06a). The processed gene expression data in this paper have been deposited into the NCBI GEO database: GSE182164. RNA-seq data download site: https://www.ncbi.nlm.nih.gov/geo/query/acc.cgi?&acc=GSE182164.

The following datasets were generated:

| Author(s) | Year | Dataset title | Dataset URL | Database and Identifier |
|---|---|---|---|---|
| Yu Z, Sun L, Zhang Y, Xia T, Lu S Li M | 2021 | Expression profiling by high throughput sequencing Non-coding RNA profiling by high throughput sequencing | https://www.ncbi.nlm.nih.gov/geo/query/acc.cgi?&acc=GSE182164 | NCBI Gene Expression Omnibus, GSE182164 |
| Yu Z, Sun L, Zhang Y, Xia T, Zhou M Li K | 2022 | mRNA seq after hypothermia treatment of crush injury of the pre-chiasmatic optic nerve | https://www.ncbi.nlm.nih.gov/geo/query/acc.cgi?acc=GSE197123 | NCBI Gene Expression Omnibus, GSE197123 |

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
