## [Editor Report]

This manuscript describes a new surgical procedure to access the optic nerve in large mammals to provide therapeutic hypothermia. Therapeutic hypothermia is a powerful idea to prevent degeneration of the nervous system following trauma or other insults. This study represents a significant step forward in the use of therapeutic hypothermia in the treatment of ocular conditions.

---

## [Decision Letter]

**Decision letter after peer review:**

Thank you for submitting your article "Cold protection allows local cryotherapy in a clinical-relevant model of traumatic optic neuropathy" for consideration by *eLife*. Your article has been reviewed by 2 peer reviewers, and the evaluation has been overseen by a Reviewing Editor and Mone Zaidi as the Senior Editor. The following individual involved in review of your submission has agreed to reveal their identity: Alfredo Martinez (Reviewer #1).

Essential revisions:

Both of the reviewers agreed that the work is well-conducted, but there are some concerns, as detailed below, that need to be addressed. Also, it is felt that conclusions regarding the immediate physiological relevance are overstated and need to be scaled back.

*Reviewer #1 (Recommendations for the authors):*

The manuscript by Zhang et al. describes a new surgical procedure to access the optic nerve in large mammals to provide therapeutic hypothermia. The study has been performed with care and represents a significant step forward in the use of therapeutic hypothermia in the treatment of ocular conditions. Nevertheless, I would like to offer some suggestions to try to improve the manuscript:

1) In the introduction, the authors may want to add some previous references about the application of therapeutic hypothermia in rodents to prevent optic nerve-induced vision losses (see, for instance, https://pubmed.ncbi.nlm.nih.gov/28761115/).

2) Once the authors have the optic nerve available, they apply a very deep hypothermia (0ºC). Previous studies have shown that a local reduction of 4-10ºC is enough to unleash the beneficial effects of therapeutic hypothermia (see, for instance, https://pubmed.ncbi.nlm.nih.gov/33897437/). There is no wonder this extremely cold treatment results in the destruction of neuron´s cytoskeleton. Moreover, this may be the reason why the protease inhibitors are needed. Authors should discuss the possibility of using warmer temperatures to induce the beneficial effects and, probably, avoid the need of using the inhibitors.

3) Authors say that therapeutic hypothermia works by reducing neuroinflammation and alleviating metabolic demand (lines 59-60). This is true but, in the last few years, different groups have been working on the hypothesis that cold-shock proteins may be involved in the beneficial effects of therapeutic hypothermia (see, for instance, https://pubmed.ncbi.nlm.nih.gov/33897437/; https://pubmed.ncbi.nlm.nih.gov/27556928/). These proteins belong to the heterogeneous nuclear ribonucleoprotein (hnRNPs) family and bind specific mRNAs, prolonging their half-life, and increasing their protein expression. Since these proteins are usually involved in survival pathways, this is an interesting mechanism to explain the benefits of hypothermia. Interestingly enough, in their transcriptomic study, authors show that the protein group with a higher gene ratio was "ribonucleoprotein complex biogenesis" (Figures S3B and S4C). This could indicate the involvement of cold-sock proteins in their model. Authors may want to add a paragraph to the discussion commenting on this issue.

4) An important piece of data that is missing is how much time passed from the moment of the crush until the application of the hypothermic treatment. The authors conclude that their treatment did not translate into a significant rescue of the FVEP or PLR, although there was some reduction in neural degeneration (lines 216-227). This lack of "complete" prevention may be related with the time that passed from the injury until the treatment. Previous studies in the rat have shown an almost complete prevention of vision loss when applying hypothermia within the first hour after injury (https://pubmed.ncbi.nlm.nih.gov/28761115/). In the same line of thought, the authors also say that "making functional regeneration is extremely challenging" (lines 350-351). I tend to agree with them. But my thought is that hypothermic treatment should be directed to prevent neuronal degeneration rather than to induce axon regeneration. Here, again, the important question is the time passed from the injury until the treatment. This point should be properly discussed.

5) I´m a little confused about the statistical analysis. The authors say (correctly) that they used normality tests to decide whether parametric or non-parametric tests should be used (lines 627-629). But then I see a lot of 2-way ANOVA tests performed with sets containing just 3 or 4 independent data. In my experience, the probability of getting normal distributions with such small numbers is rather unlikely. I would appreciate if the authors could confirm that all the experiments where they used parametric tests had been confirmed as having a normal distribution. Otherwise, non-parametric tests should be performed.

*Reviewer #2 (Recommendations for the authors):*

Beyond the critiques in the Public Comments section, I have the following:

1. The Introduction (particularly lines 86 – 96) summarize the experimental results too much rather than just introducing the experimental questions and approach. If anything, this summary would be better suited for the Discussion.

2. In Figure 1, the term "sham" is used in describing the surgery to expose the prechiasmal optic nerve without injuring it. This term is not used in the text of the Results section, and so it is confusing to the reader. (For instance, "sham" might refer to just putting the endoscope into the nasal cavity without penetrating bone). It should be made clear that sham is doing the entire procedure and just stopping short of intentionally crushing the nerve.

3. In multiple places in the figure legends, the RGC marker RBPMS is mistakenly written as "RBMPS".

4. Figure 2A: to increase understanding, the right-most panel should show some compressed optic nerve tissue being pressed against the remaining wall of the optic canal. The fact that no optic nerve is shown at all in this panel is confusing (as if it has been removed).

5. Figure 2B: are the two panels at different magnifications? Should adjust so that they are the same magnification or provide a scale bar.

6. Figure 2C: the placement of asterisks at PLR = 1 is confusing, because these look like data points. I believe that the actual data is a PLR of 0 for all post-injury time points (shown as colored line segments that are easy to overlook).

7. Figure 3A: Are the authors surprised that even at 3 months post-injury the GCC thickness is >70% that of the contralateral eye? In humans, I would expect much more thinning than this by 3 months. Furthermore, the macaque model shows much more GCC thinning at this time point. Is this a species-specific effect?

8. Figure 3A: Why is there a dip in GCC thickness in the inferonasal region (NI) at all time points (including baseline)? This was not observed in Figure 1I.

9. Figure 5I: looking at the y-axis (where the lowest value is 80% rather than 0%), it is clear that there is not much true difference between the GCC thickness of any of these conditions, although it looks like the p-value was slightly below 0.05. If any argument is going to be made about RGC cell body protection, then I think retinal flat mounts stained with RBPMS for soma counting (as in Figure 1J or 3B) would be essential.

10. The transcriptional profiling in Figure 4 has very little relation to the rest of the paper. If this were compared to optic nerve mRNA after optic nerve crush and hypothermia/PI treatment and significant changes between the two conditions were noted, this would be much more interesting.

---

## [Author Response]

Reviewer #1 (Recommendations for the authors):The manuscript by Zhang et al. describes a new surgical procedure to access the optic nerve in large mammals to provide therapeutic hypothermia. The study has been performed with care and represents a significant step forward in the use of therapeutic hypothermia in the treatment of ocular conditions. Nevertheless, I would like to offer some suggestions to try to improve the manuscript:1) In the introduction, the authors may want to add some previous references about the application of therapeutic hypothermia in rodents to prevent optic nerve-induced vision losses (see, for instance, https://pubmed.ncbi.nlm.nih.gov/28761115/).

We appreciate Reviewer 1’s valuable suggestion regarding this key reference. The reference found that hypothermia was an efficacious treatment for traumatic vision-impairing conditions and that the cold-shock protein pathway may be involved in mediating the beneficial effects.

We added this reference in the first paragraph of the Introduction, as well as in the Discussion section.

2) Once the authors have the optic nerve available, they apply a very deep hypothermia (0ºC). Previous studies have shown that a local reduction of 4-10ºC is enough to unleash the beneficial effects of therapeutic hypothermia (see, for instance, https://pubmed.ncbi.nlm.nih.gov/33897437/). There is no wonder this extremely cold treatment results in the destruction of neuron´s cytoskeleton. Moreover, this may be the reason why the protease inhibitors are needed. Authors should discuss the possibility of using warmer temperatures to induce the beneficial effects and, probably, avoid the need of using the inhibitors.

We appreciate Reviewer 1 pointing out the importance of the temperature control in hypothermia treatment. We agree that a local reduction of 4-10ºC is enough to unleash the beneficial effects of therapeutic hypothermia. In the ex vivo experiment, we did apply deep hypothermia (0 ºC), but in the in vivo experiment, we applied 4 ºC, but not 0 ºC, local hypothermia to the injured optic nerve. Figure 5E, 5G shows that, although the water temperature in the tank was 0 ºC, the surface temperature of the cling film, which contacted the sponge over the injury site, was measured to be 4 ºC.

We apologize for the confusion and stress this point in the Results section. Nonetheless, because, in our previous study, we found that 4 ºC deep hypothermia will cause neuronal skeleton dysfunction, we added protease inhibitors as hibernation-mimicking cold protection in the present study. We have added more discussion about the need to optimize the hypothermia therapy temperature. For example, in the Discussion we now say, “Warmer temperatures in between 4-10ºC may achieve neuroprotection without damaging cytoskeleton, which needs to be further optimized through in vitro and in vivo studies.”

3) Authors say that therapeutic hypothermia works by reducing neuroinflammation and alleviating metabolic demand (lines 59-60). This is true but, in the last few years, different groups have been working on the hypothesis that cold-shock proteins may be involved in the beneficial effects of therapeutic hypothermia (see, for instance, https://pubmed.ncbi.nlm.nih.gov/33897437/; https://pubmed.ncbi.nlm.nih.gov/27556928/). These proteins belong to the heterogeneous nuclear ribonucleoprotein (hnRNPs) family and bind specific mRNAs, prolonging their half-life, and increasing their protein expression. Since these proteins are usually involved in survival pathways, this is an interesting mechanism to explain the benefits of hypothermia. Interestingly enough, in their transcriptomic study, authors show that the protein group with a higher gene ratio was "ribonucleoprotein complex biogenesis" (Figures S3B and S4C). This could indicate the involvement of cold-sock proteins in their model. Authors may want to add a paragraph to the discussion commenting on this issue.

We appreciate the reviewer’s insightful suggestion. We have checked the mRNA expression of RNA-binding motif protein 3 (RBM3) and cold inducible RNA-binding protein (CIRP) at day 1 post hyperthermia treatment in our Crush injury model. We found that they were not elevated compared to the non-treatment optic nerve. Their mRNA expression changes were very stable: their log2 Fold Change was -0.0178 and 0.0064, respectively. A potential explanation is that some cold-shock proteins such as RMB3 are more involved when hypothermia is mild (30-33C, via a systemic hypothermia approach) but not in deep hyperthermia (4C), as documented in the literature (PMID: 23415676; PMID: 27147467)

**Author response table 1. sa2table1:** 

Potential differences	Previous studies (PMID: 33897437; 27556928)	This study
Delivery of hypothermia	Systemic hypothermia	Local hypothermia
Temperature of hypothermia	Mild (30-33C)	Deep (4C)
Species	Rodent	Goat
Sampling tissue	Retina (soma of RGCs, where the nucleus and most ribosomes lie)	Pre-chiasmatic optic nerve (distal axons of RGCs)

To address the reviewer’s suggestion, we have added the following discussion and cited these two references (PMID: 33897437; 27556928) in the revised manuscript:

“Previous studies also indicated that hypothermia-induced elevated expression of cold-shock proteins may contribute to neuronal protection^34，35^, especially in the mild hypothermia range ^36,37^.”

4) An important piece of data that is missing is how much time passed from the moment of the crush until the application of the hypothermic treatment. The authors conclude that their treatment did not translate into a significant rescue of the FVEP or PLR, although there was some reduction in neural degeneration (lines 216-227). This lack of "complete" prevention may be related with the time that passed from the injury until the treatment. Previous studies in the rat have shown an almost complete prevention of vision loss when applying hypothermia within the first hour after injury (https://pubmed.ncbi.nlm.nih.gov/28761115/). In the same line of thought, the authors also say that "making functional regeneration is extremely challenging" (lines 350-351). I tend to agree with them. But my thought is that hypothermic treatment should be directed to prevent neuronal degeneration rather than to induce axon regeneration. Here, again, the important question is the time passed from the injury until the treatment. This point should be properly discussed.

We agree with the reviewer that the timing of TH treatment is essential. To address the reviewer’s concern, we have added the specific time information to the revised manuscript, as follows: “Immediately after the ON crush injury, the sponge was placed at the injury site and then the local cooling tubes were inserted over the gelfoam sponge. Local hypothermia started within 20 minutes after optic nerve crush.”

We also agree with the reviewer that hypothermic treatment should prevent neuronal degeneration rather than induce axon regeneration. We have added the following discussion in the revised manuscript: “Hypothermic treatment may directly prevent neuronal degeneration, and early treatment is necessary to alleviate further progression of neural damage after optic nerve injury (PMID: 28761115), as well as after other CNS trauma (PMID: 28629741, PMID: 11765843).”

5) I´m a little confused about the statistical analysis. The authors say (correctly) that they used normality tests to decide whether parametric or non-parametric tests should be used (lines 627-629). But then I see a lot of 2-way ANOVA tests performed with sets containing just 3 or 4 independent data. In my experience, the probability of getting normal distributions with such small numbers is rather unlikely. I would appreciate if the authors could confirm that all the experiments where they used parametric tests had been confirmed as having a normal distribution. Otherwise, non-parametric tests should be performed.

We used residual analysis to check the assumption of normal distribution of the raw data before 2-way ANOVA analysis, because it is believed to be a better way to check ANOVA assumptions than statistical tests (Kozak, M. & Piepho, H.-P. What's normal anyway? Residual plots are more telling than significance tests when checking ANOVA assumptions. Journal of Agronomy and Crop Science 204, 86-98 (2018)).

Residuals are the differences between the raw data and the estimated values; residual normality was tested using the Shapiro-Wilk test or Kolmogorov-Smirnov test in GraphPad (9.0) software. Of note, residual normality is tested automatically when performing two-way ANOVA test in the GraphPad (9.0) software.

To make our approach clearer, we have added the following statement in the Methods section: “Residual analysis was used to check the assumption of normal distribution of the raw data before 2-way ANOVA analysis, because it has been shown to be a better way to check ANOVA assumptions than statistical tests50.”

We thank the reviewer for his reminding. When we re-checked the statistics, we found that the raw data in original Figure 3C did not pass residual normality test before two-way ANOVA test. Therefore, we used unpaired t-test or Mann-Whitney test instead to compare the axon densities between 1mpi and 3mpi at different optic nerve sites (#1, 2, 3, 4).

In the revised manuscript, we confirm that all the results in which parametric tests had been used had a normal distribution. The result of the normality test for each analysis in this manuscript is available in the supplementary file named “statistics.xlsx”.

Reviewer #2 (Recommendations for the authors):Beyond the critiques in the Public Comments section, I have the following:1. The Introduction (particularly lines 86 – 96) summarize the experimental results too much rather than just introducing the experimental questions and approach. If anything, this summary would be better suited for the Discussion.

We appreciate the reviewer’s constructive comments and have simplified these experimental results in the Introduction.

2. In Figure 1, the term "sham" is used in describing the surgery to expose the prechiasmal optic nerve without injuring it. This term is not used in the text of the Results section, and so it is confusing to the reader. (For instance, "sham" might refer to just putting the endoscope into the nasal cavity without penetrating bone). It should be made clear that sham is doing the entire procedure and just stopping short of intentionally crushing the nerve.

We agree. To make it clear, we state in the Results section that “For goats in the sham group, the ON was exposed without crush.” We also mention in the Methods section that “In the sham surgery group, the ON was exposed by drilling off its anterior bony wall without ON crush (Figure 1G)”.

3. In multiple places in the figure legends, the RGC marker RBPMS is mistakenly written as "RBMPS".

We are indebted to the reviewer for his careful review. We have corrected “RBMPS” into “RBPMS” in Figure 1J and 3B.

4. Figure 2A: to increase understanding, the right-most panel should show some compressed optic nerve tissue being pressed against the remaining wall of the optic canal. The fact that no optic nerve is shown at all in this panel is confusing (as if it has been removed).

This is a great suggestion and we have modified Figure 2A accordingly.

5. Figure 2B: are the two panels at different magnifications? Should adjust so that they are the same magnification or provide a scale bar.

We updated the two panels in Figure 2B at the same magnification.

6. Figure 2C: the placement of asterisks at PLR = 1 is confusing, because these look like data points. I believe that the actual data is a PLR of 0 for all post-injury time points (shown as colored line segments that are easy to overlook).

Yes, the actual data is a PLR of 0 for all post-injury time points. We have changed the placement of asterisks.

7. Figure 3A: Are the authors surprised that even at 3 months post-injury the GCC thickness is >70% that of the contralateral eye? In humans, I would expect much more thinning than this by 3 months. Furthermore, the macaque model shows much more GCC thinning at this time point. Is this a species-specific effect?

We thank the reviewer for pointing out the contrast between our model and human patients in the time course of progressive GCC thinning. According to a small case series report in human patients with TON (PMID: 22893082), the final status of GCC at 20 months after injury was 64%, 68%, 71%, and 74% of the contralateral eye in cases 1 through 4, respectively. Therefore, at least in this small series, human patients did not experience more severe GCC thinning than we found in our goat model at 3 months. Many other TON clinical studies focus on retinal nerve fiber layer (RNFL) thinning. However, the boundary between RNFL and ganglion cell layer became fuzzy in goat after optic nerve injury. Therefore, we did not quantify RNFL in this study.

The macaque model shows much more GCC thinning (<50%) at 3 months after injury than the goat model, likely due to much higher retinal ganglion cell density in the macaque (~1000 cells/1 mm^2, Figure S7) than in the goat (~200 cells/ 1mm^2, Figure 5A, B in our previous study (PMID: 32598972)).

To address the reviewer’s comment, we have added the following discussion in the revised manuscript: “Of note, the macaque model shows much more GCC thinning (<50%) at 3 months after injury than the goat model, likely due to much higher retinal ganglion cell density in the macaque (~1000 cells/1 mm^2, Figure S7) than the goat (~200 cells/ 1mm^2, (PMID: 32598972)”).

8. Figure 3A: Why is there a dip in GCC thickness in the inferonasal region (NI) at all time points (including baseline)? This was not observed in Figure 1I.

Figure 1 and 3 showed the OCT thickness of two different groups of goats. The dip in GCC thickness in the NI region indicated that for these goats in the optic nerve crush group (Figure 3), the surgery eyes had thinner GCC in the NI region than their contralateral eyes (control eyes) at baseline. In the sham group (Figure 1), the surgery eyes had thinner GCC in the NS regions and thicker GCC in the TS region at baseline.

These regional differences in GCC thickness between ipsilateral and contralateral eyes at baseline will not affect our results because our study is based on longitudinal observations.

9. Figure 5I: looking at the y-axis (where the lowest value is 80% rather than 0%), it is clear that there is not much true difference between the GCC thickness of any of these conditions, although it looks like the p-value was slightly below 0.05. If any argument is going to be made about RGC cell body protection, then I think retinal flat mounts stained with RBPMS for soma counting (as in Figure 1J or 3B) would be essential.

In the control group without treatment, the average GCC thickness ratio of the injured eyes at 1 mpi was 88% of the baseline ratio. In other words, 12% GCC thickness was lost at 1 mpi.

In the hypothermia+PI group, the GCC thickness ratio of the injured eyes at 1 mpi was 91% of the baseline ratio, so 9% GCC thickness was lost at 1mpi.

Therefore, 25% (25% = (12%-9%)/12%) of GCC thickness was saved at 1 mpi due to local treatment with hypothermia/PI. The therapeutic effect, although not substantial, was encouraging evidence for the potential of local treatment.

As shown in Figure 3B, RGCs density ratio of the injury eye to the contralateral control eye was almost 100% at 1 mpi in the no treatment group (little RGC loss at 1 mpi). Therefore, we did not carry out RGCs density analysis between the no treatment group and the hypothermia group at 1 mpi.

10. The transcriptional profiling in Figure 4 has very little relation to the rest of the paper. If this were compared to optic nerve mRNA after optic nerve crush and hypothermia/PI treatment and significant changes between the two conditions were noted, this would be much more interesting.

Transcriptional profiling in Figure 4 revealed early mRNA expression changes at different sites in our TON model, helping us understand the pathophysiological process and identify potential therapeutic targets.

Transcriptomic analysis in Figure 4 revealed early changes in ischemia, inflammation, and metabolic pathways in the injured optic nerve, justifying local hypothermia as a therapeutic candidate. Therefore, we applied local hypothermic therapy in our model (Figure 5).

We do agree with the reviewer that it is interesting to compare mRNA changes between no treatment and hypothermia/PI treatment. We have performed this comparison and added the new data as Figure 6 in the revised manuscript.

We have included the following results in the revised manuscript: “To explore transcriptomic change after local treatment with deep hypothermia and PI, we sampled from the injured pre-chiasmatic ON segments at 1 day post injury (dpi) and compared transcriptomic expression between the hypothermia/PI treatment group and no treatment group. There were 264 differentially expressed genes after hypothermia/PI treatment, of which 220 genes were upregulated and 44 genes were downregulated (Figure 6A, B). Gene list analysis by Metascape revealed that many inflammatory, immune-related functional cluster groups of gene ontology terms were enriched, such as neutrophil degranulation, inflammatory response, and regulation of cell adhesion (Figure 6C). Scored pro-inflammatory response (GO:0050729: positive regulation of inflammatory response) and anti-inflammatory response (GO:0050728: negative regulation of inflammatory response) by GSEA revealed that the enrichment score of negative regulation of inflammation was higher than that of positive regulation, indicating that hypothermia/PI treatment negatively regulated inflammatory response overall (Figure 6D). SsGSEA analysis (Figure 6E) found that the anti-inflammatory response was significantly improved after hypothermia treatment. QuanTIseq analysis, which was used to predict cell types within each sample, found a significant decrease in the number of neutrophils and a significant increase in the number of B cells, T cells and mDCs (myeloid dendritic cells), indicating that innate immune response was suppressed while adaptive immune response was enhanced after hypothermia/PI treatment (Figure 6F). These results need to be confirmed by single cell sequencing in the future study.”